# ARE LARGE REASONING MODELS INTERRUPTIBLE?

## ABSTRACT

Large Reasoning Models (LRMs) excel at complex reasoning but are traditionally evaluated in static, "frozen world" settings: model responses are assumed to be instantaneous, and the context of a request is presumed to be immutable over the duration of the response. While generally true for short-term tasks, the "frozen world" assumption breaks down in modern reasoning tasks such as assistive programming, where models may take hours to think through problems and code may change dramatically from the time the model starts thinking to the model's final output. In this work, we challenge the frozen world assumption and evaluate LRM robustness under two realistic dynamic scenarios: interruptions, which test the quality of the model's partial outputs on a limited budget, and dynamic context, which tests model adaptation to in-flight changes. Across mathematics and programming benchmarks that require long-form reasoning, static evaluations consistently overestimate robustness: even state-of-the-art LRMs, which achieve high accuracy in ideal settings, can fail unpredictably when interrupted or exposed to changing context, with performance dropping by up to 60% when updates are introduced late in the reasoning trace. Our analysis further reveals several novel failure modes, including *reasoning leakage*, where models fold the reasoning into their final answer when interrupted; *self-doubt*, where performance degrades while incorporating update information; and *panic*, where under time pressure models abandon reasoning entirely and return incorrect answers.

## 1 INTRODUCTION

Large Reasoning Models (LRMs) have achieved state-of-the-art performance on complex, multi-step reasoning tasks (Agarwal et al., 2025; Yang et al., 2025). The dominant paradigm for evaluating these models, however, remains static and turn-based. In this setup, a model receives a fixed problem, generates a complete response, and the environment is assumed to be "frozen" during its computation. By adopting this sequential binary interaction paradigm, existing models fail to capture the fluid and interactive nature of real-world problem-solving, where environments evolve, collaborators intervene, and goals change.

For example, a user may want to interrupt a long-running computation (test-time compute) to get a quick, partial answer. Perhaps the user observes a flaw in the reasoning or their initial request and would like to interject new information or instructions. Alternatively, a coding agent needs to be able to work in an environment where other agents and users are also modifying the same codebase. The standard method for handling these intervening situations – aborting the entire process, manually editing the context, and restarting from scratch – is inefficient, disrupts user workflows, and loses important partial context.

Intuitively, interrupting a generation seems as simple as closing the reasoning or agent message and inserting user feedback. Indeed, some public models already expose ad-hoc mechanisms (e.g., injecting a new user turn, or discarding existing traces and restarting). However, what are the implications of these modes of intervention? Do LRMs produce progressively better answers as more of the reasoning trace accrues, and how gracefully do they degrade under hard stops? Can models heed requests to speed up by compressing or truncating their own reasoning, without loss in quality? When new information arrives mid-inference, can models recognize and respond to the problem shift? And, how sensitive are these behaviors to the timing and surface form of the intervention?

In this work, we investigate directly how LRMs perform under these realistic time-sensitive and dynamic conditions (Figure 1). We focus on two primary types of dynamic intervention. The first is **interruptions**, where a model's reasoning is cut short and it must produce the most coherent and helpful answer possible within its allotted computational budget. The second is **dynamic context**, where the problem's specifications or environment change mid-inference, requiring the model to detect, integrate, and adapt

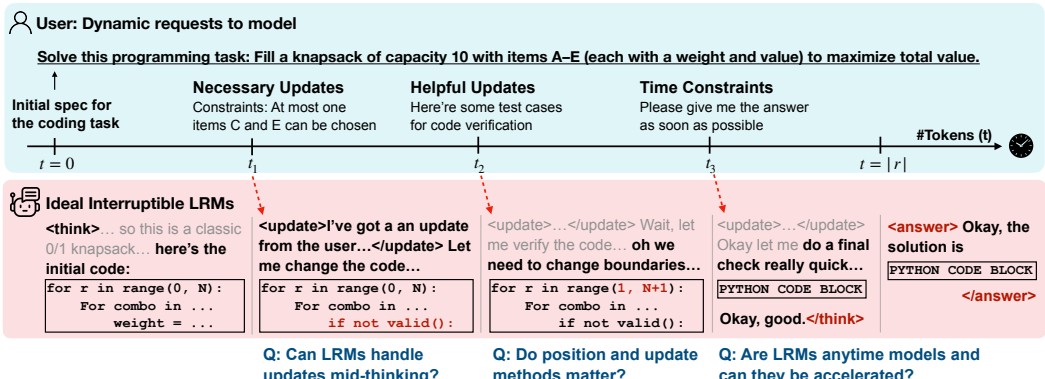

Figure 1: **How do LRMs perform in dynamic worlds?** In contrast to static settings, where users can only wait for completion or terminate generation, since LRM reasoning can be time-consuming, real-world use cases often require mid-inference updates. We introduce a new public dataset and benchmark with novel evaluation protocols for assessing model performance under interruption and dynamic context updates across tasks in mathematics and coding, showing that while models have an approximately any-time behavior, they are far from robust across all dynamic scenarios.

to the new context. By analyzing model behaviors in these settings, we aim to understand and characterize their limits and failure points when the "frozen world" assumption no longer holds.

To perform these analyses, we introduce a suite of evaluation protocols for dynamic environments built upon benchmarks in two domains, math (Lightman et al., 2024; Cobbe et al., 2021; Codeforces, 2024) (GSM-8K, Math-500, AIME) and programming (Jain et al., 2024) (LiveCodeBench), that require long-form reasoning. In the case of interrupt signals, we measure whether models adhere to stop signals and evaluate the quality of their partial outputs. For interruptions that contain new information, we evaluate how effectively the new information is integrated into the models' reasoning process and final answers.

Overall, we find that static evaluations, while useful, consistently overestimate the practical robustness of modern LRMs. We show that even top-performing models, which achieve high accuracy in ideal settings, can fail unpredictably when interrupted or presented with new information mid-task, with accuracy dropping by up to 60% when updates are provided late in the reasoning trace, after models have committed to an answer. We also find several interesting secondary effects, including a "reasoning-leakage" effect, where models think within the answer of the problem when interrupted during the thinking process, model "self-doubt," leading to performance degradation while incorporating updated information, and model "panic" where, when requested to hurry up, models abandon reasoning, and immediately return incorrect answers.

Our contributions are thus threefold:

- We propose a new analytical framework for evaluating LRMs as persistent, interruptible agents operating in dynamic environments.
- We introduce a new public dataset and benchmark with novel evaluation protocols for assessing model performance under interruption and dynamic context updates across tasks in mathematics and coding.
- We provide an empirical analysis that identifies and characterizes common failure modes in state-of-the-art models, and identify several interesting downstream effects of interruption on model performance and robustness.

## 2 BACKGROUND & RELATED WORK

LRMs such as OpenAI's O1/O3/O4, Gemini, DeepSeek (Guo et al., 2025), Qwen3 (Yang et al., 2025), and others have pushed the boundaries of AI problem-solving by leveraging extended chain-of-thought reasoning. These models generate explicit step-by-step reasoning sequences that can help the model solve complex tasks in domains such as mathematics and programming. Longer and more detailed reasoning paths often correlate with higher accuracy (Guo et al., 2025; Yang et al., 2025; Agarwal et al., 2025; Cheng et al., 2025; Bi et al., 2025). These models are traditionally evaluated on a suite of complex tasks, including math (Lightman et al., 2024; Cobbe et al., 2021; Codeforces, 2024), programming (Jain et al., 2024; Jimenez et al., 2023), and question answering (Rein et al., 2024; He et al., 2024).

Unfortunately, the benefits of extended reasoning are often accompanied by significantly increased computational costs and latency. In the extreme, reasoning can lead to "overthinking" in which the reasoning model produces excessively verbose and redundant outputs, even for simple queries (Sui et al., 2025a). This inefficiency has led to a large body of research on "efficient reasoning," which aims to optimize reasoning length while maintaining accuracy. Methods for efficient reasoning broadly take two forms: training-based methods (Yu et al., 2024; Xia et al., 2025; Arora & Zanette, 2025; Liu et al., 2024; Munkhbat et al., 2025), and inference-time interventions (Muennighoff et al., 2025; Ma et al., 2025; Nayab et al., 2024; Xu et al., 2025; Yan et al., 2025) designed to control reasoning without modifying the model itself.

In this work, we focus primarily on inference-time interventions in reasoning models. One of the most direct inference-time approaches, NoThinking (Ma et al., 2025), questions the necessity of the explicit reasoning phase altogether and uses a simple prompt to bypass the "thinking" block and proceed directly to the final solution. In this work, we take this idea a step further by probing models to stop thinking at different points, and we find that LRMs often continue reasoning *outside* the dedicated `<think>` tokens, a behavior we call "reasoning leakage."

Budget forcing (Muennighoff et al., 2025) studies models under hard cutoffs imposed by fixed token or step budgets. Their findings highlight issues such as overshooting step limits and the positive correlation between the number of reasoning steps and accuracy. The setup in Muennighoff et al. (2025) is similar to our "interrupt" setting; however, we take a different view of the performance: our notion of hard interruption arises in interactive, dynamic settings, where the cutoff is externally imposed by the user or environment, often unpredictably. We thus evaluate the robustness of the final answer under early termination and under interruptions that may also inject new or adversarial instructions, rather than explore how a pre-determined budget can be met.

Beyond these simple approaches, more nuanced control mechanisms have also been proposed, including TALE Han et al. (2024), Budget Guidance (Li et al., 2025b), Sketch-Of-thought (Aytes et al., 2025), ThinkLess (Li et al., 2025a), NoWait (Wang et al., 2025), Chain of Draft (Xu et al., 2025), Constrained-CoT (Nayab et al., 2024), Meta-Reasoner (Sui et al., 2025b), and Inftythink (Yan et al., 2025). These studies universally operate under a static "frozen world" assumption: a problem is presented to the model, and its context is presumed to be immutable throughout the reasoning process. In contrast, our work is not concerned with bypassing or limiting the thinking process, but rather with evaluating its resilience and the utility of its partial outputs when faced with interruption.

Perhaps closest to our work is Fan et al. (2025), who show that the overthinking phenomenon is exacerbated by missing premises in the questions, which makes those questions "unsolvable" in their initial state. They find that in these scenarios, reasoning models, rather than identifying the missing information and abstaining, generate drastically longer responses (2-4x more tokens) in a futile attempt to find a solution. They further show that models often express suspicion about a missing premise early in their reasoning process but lack the confidence to terminate, instead falling into repetitive loops of self-doubt and hypothesis generation. Our paper expands on these ideas, by exploring how models incorporate updates to the context which fill these missing premises, or correct for misconceptions in the original context.

## 3 HOW TO (POLITELY) INTERRUPT A MODEL?

LRMs typically generate a reasoning trace before producing a final answer. In interactive environments, however, users may wish to obtain an early answer or provide new information mid-inference. To handle such settings, LRMs must be robust to *interruptions* during their thinking process. In this work, we focus on two complementary classes: *time-constrained*, where the user accelerates output by limiting the reasoning budget, and *update-driven*, where new information modifies the problem specification. We first formalize this problem setup, and then describe the scenarios and evaluation dimensions for each setting.

### 3.1 PROBLEM SETUP

**Model Inference.** Let $M$ denote an autoregressive LRM and $q$ a query. Under a static setting, the model takes $q$ and outputs a reasoning trace $r = (r_1, r_2, ..., r_T)$ of token length $T$ together with a final answer $a$:

$$M(q) \mapsto (r, a)$$

Under standard static evaluation settings, we measure the correctness based on the final answer $a$.

However, as mentioned earlier, we aim to evaluate models under dynamic environment settings, where interruptions are introduced during the model's thinking process. Formally, instead of letting the model generate the full reasoning trace, $r$, we stop the generation at a given point, $X$, and insert interruption tokens, $i$. Specifically, the inference is broken down into two stages:

$$(1)\ M(q) \mapsto (r_{:X}), \qquad (2)\ M(q, r_{:X}, i) \mapsto (r'_{X:}, a').$$

In the first stage, the model, $M$, generates its reasoning trace up to a given point, $X$, $r_{:X} = (r_1, ..., r_{\lfloor XT \rfloor})$. In the second stage, the model, $M$ is conditioned on its prior input query ($q$), its interrupted reasoning trace ($r_{:X}$), and interruption tokens ($i$), outputting its remaining reasoning trace, $r'_{X:}$, and a final answer, $a'$.

It should be noted that the above setup can be adapted to multiple interrupts, breaking down the model's inference process into more than two stages.

**Evaluation.** We evaluate two aspects: **correctness** and **length**, which serves as a proxy for inference computation. For correctness, we define the interruption-conditioned accuracy

$$A_i(X) \triangleq \Pr[a' = a^* \mid X, i],$$

where $a'$ is the output after interruption, $a^*$ the interruption-aware ground truth, $X$ the cut point, and $i$ the interruption tokens. This extends static accuracy to dynamic settings. For length, we measure the number of tokens generated after interruption,

$$L_i(X) = |r'_{X:} \oplus a'|,$$

which captures the computation cost of producing $a'$. For comparison, the static (no-interruption) cost is

$$L^*(X) = |r_{X:} \oplus a|.$$

### 3.2 Interrupt Scenarios

With the problem setup in place, we now turn to the concrete scenarios that motivate our study. Interruptions can arise in two distinct ways: (i) time-constrained, when the user imposes a deadline or requests faster responses, and (ii) update-driven, when the task specification itself changes during the reasoning process.

**Time-constraint interruptions.** Here, the user requests acceleration and interrupts at step $X$. In a *hard interrupt* scenario, the user injects $i \in \{\langle \text{end-thinking} \rangle, \langle \text{force-answer} \rangle\}$, so the model is prompted with $(q, r_{:X}, i)$ and is forced to terminate its reasoning:

$$M(q, r_{:X}, i) \mapsto (r'_{X:} = \emptyset, a').$$

The token $\langle \text{end-thinking} \rangle$ simply closes the reasoning block, whereas $\langle \text{force-answer} \rangle = \langle \text{end-thinking} \rangle + \delta$ requires immediate output in a prescribed format (e.g., $\delta =$ "\boxed{" in math or code fences in programming). In the first case, the model may emit a minimal chain-of-thought before $a'$, while in the second case it must directly output $a'$. This setting allows us to probe (i) whether $A_i(X)$ is approximately non-decreasing in $X$ (anytime behavior) and (ii) whether the two termination signals induce different behaviors.

In the *soft interrupt* scenario, the tokens $i$ is a directive (e.g., "Please answer faster"), and the model is not forced to stop reasoning:

$$M(q, r_{:X}, i) \mapsto (r'_{X:}, a'), \quad r'_{X:} \neq \emptyset.$$

Here, the model continues generating $r'_{X:}$ but may have its reasoning dynamics altered due to the instruction. Such directives can lead the model to compress its reasoning, reduce verbosity, or terminate early to provide an answer. We study whether $L_i(X)$ is shorter than $L^*$ and how this affects accuracy, as well as how the locus of intervention matters: a directive given as a user-turn message may be interpreted differently than a control token injected into the reasoning context; timing (earlier vs. later in the trace) may also influence outcomes.

**Update-driven interruptions.** In this scenario, the interruption conveys new information that modifies or helps with the task. Formally, we prompt the model with $(q, r_{:X}, u)$, where $u$ encodes the update content:

$$M(q, r_{:X}, u) \mapsto (r'_{X:}, a').$$

Let $a^*(q)$ denote the ground-truth answer under the original query and $a^*(q, u)$ under the updated query. A necessary update satisfies

$$a^*(q) \neq a^*(q, u),$$

so that incorporating $u$ is required to produce a correct answer. In this work, we focus on single, useful updates, although the framework naturally extends to multiple and other types of updates (e.g., distracting).

To construct such update-driven interruptions, we augment standard reasoning datasets in math and programming, including GSM-8k, MATH500, AIME24/25, and LiveCodeBench-v6. For math tasks, we modify initial conditions (for example, changing variable values) so that the updated problem $p'$ together with $u$ is semantically equivalent to the original problem $p$. For programming tasks, we first provide only the textual problem description, then introduce updates $u$ that alter starter code, adjust variable ranges, or add constraints and sample test cases. All augmentations are generated with GPT-5 and manually verified by the authors, ensuring that $u$ is required for correctly solving the problems. An example of updates and additional details on the dataset construction are provided in Appendix E.

### 3.3 Experimental Design

We evaluate state-of-the-art LRMs on math and coding tasks under both the time-constrained and update-driven interruption settings. Specifically, we consider Qwen3-8B (Yang et al., 2025), GPT-OSS-20B (high reasoning effort) (Agarwal et al., 2025), and Magistral-Small-1.2 (Rastogi et al., 2025) as three diverse and representative models. For interruption positions, denoted as $X$, we avoid using an absolute token threshold since reasoning lengths vary significantly across models and even across samples. Instead, we define $X$ as a relative fraction of the full reasoning trace length $R_T$. For each sample, we first obtain the full reasoning trace and then simulate interruptions at $X \in \{0.1, 0.3, ..., 0.9\} \cdot R_T$.

We evaluate on both math and coding benchmarks. The math tasks include a 500-example subsample of GSM8K (without in-context examples), MATH-500, and AIME24/25. For coding, we use LiveCodeBench-v6, filtering problems released after October 1st, 2024, following the same setup as Qwen3. In the time-constrained setting, we directly run inference on the official datasets, whereas in the update-driven setting, we use the augmented datasets described in the previous section. Following the DeepSeek-R1 (Guo et al., 2025) evaluation protocol, we run 16 independent trials for AIME24/25 due to its small size and high variance, and a single run for the other datasets. We report the mean accuracy along with bootstrapped 95% confidence intervals. All experiments are conducted using the vLLM framework on NVIDIA Ampere or newer GPUs, depending on model size; see Appendix C for more details. In the next two sections, we mainly focus on the mentioned four tasks, three models, and two interruption scenarios. Additional analyses and results, including the discussions on Deepseek-R1, are in Appendix F.

## 4 How do models behave under time-constraint interruptions?

One of the most common types of interruption, and the first that we investigate, is the explicit termination of the thinking process; the "**hard interrupt**" scenario discussed in subsection 3.2. This reflects an "anytime" scenario in which the user wants the model to produce an answer without further delay. The overall accuracy for this experiment is shown in Figure 2 (top). In both math and programming problems, we see anytime behavior: if we interrupt models earlier in the thinking process, the performance is worse than if we interrupt them late in the thinking process. This holds in all cases except for Magistral on coding tasks, where interrupting late in the process leads to slightly improved performance relative to not interrupting.

Interestingly, however, when we look at the overall length of the reasoning traces after interruption, shown in Figure 2 (bottom), we see something surprising: in the hard interrupt scenario, the final answer length is the same as or exceeds the length of the final answer generated under full thinking settings. This is due to an effect we call "reasoning leakage," where instead of stopping the thinking process, the model continues to think in the answer portion of the text before providing the final answer. To explore this further, we run a harder version of the interrupt process, which we call, "**extreme hard interrupt**," where we force the model to produce an answer to the question, either by inserting the markdown code indicator ``` for coding questions, or by inserting the `box` indicator for the math questions. Here, we observe that for math problems, this dramatically reduces both the accuracy and answer length, as the model is forced to provide an immediate answer, and no reasoning leakage occurs. For coding problems, on the other hand, this scenario has no effect, as the model continues reasoning *in the comments of the answer code*, as seen in Listing D.1 in the appendix.

We also explore how models react to soft interrupts, where they are instructed to accelerate their reasoning but are still allowed to continue their thought process, the "**soft interrupt**" scenario as discussed in subsection 3.2. Figure 3 (top) shows that on average, models perform quite well under soft interrupts at any point in the thinking process, and it is only on hard tasks such as AIME for GPT-OSS where soft-interrupts make a significant impact on the overall performance of the model (where GPT-OSS "panics" and outputs

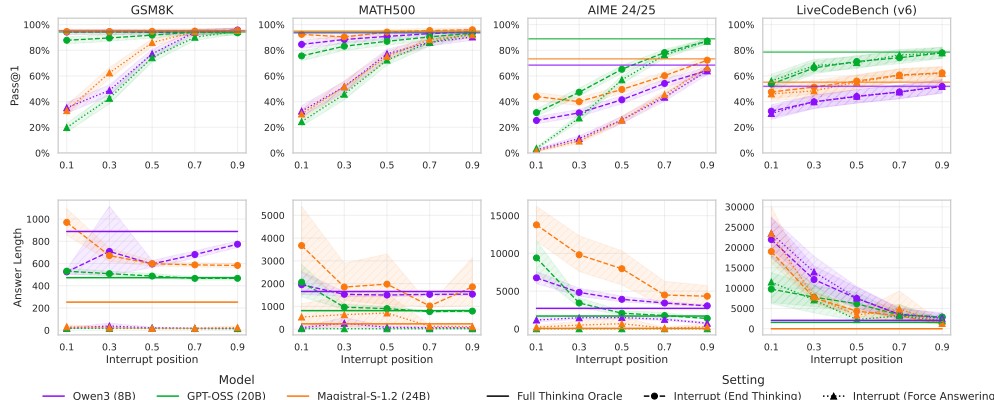

Figure 2: **Efficiency and Accuracy for Hard Interrupts.** The top row reports model performance (Pass@1, denoted as $A(X)$ in Section 3), while the bottom row shows absolute final answer lengths $L(X)$ under two settings across different interrupt position X. In the top row, we observe that LRMs behave almost like anytime models, with performance improving as more reasoning budget is provided. In the bottom row, we find evidence of *reasoning leakage*: when interrupted too early, models often continue reasoning in their final answers despite being forcibly terminated.

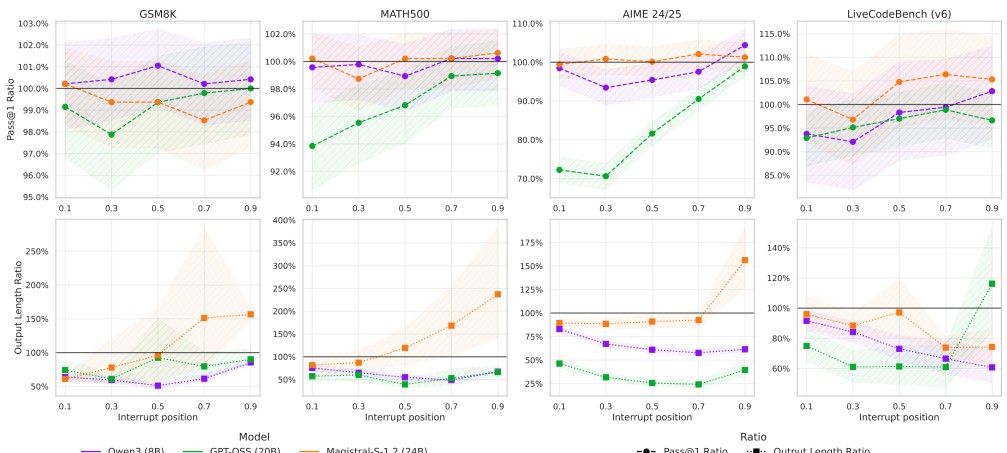

Figure 3: **Efficiency and Accuracy for Soft Interrupts.** The top row shows the accuracy ratio compared to a full-thinking model, with the bottom row showing the output length ratio compared to a full-thinking model. Soft interrupts significantly reduce the tokens used in the remaining output but have little impact on overall accuracy, except in hard tasks, where GPT-OSS suffers from "answer panic", and immediately outputs incorrect answers.

an incorrect answer, often immediately). A qualitative example is shown in subsection D.2. Overall, models generally obey the soft prompt and indeed produce fewer tokens than a non-interrupted model. Figure 3 (bottom) shows that while soft prompting late in the thinking trace for Magistral increases prompt length, soft prompting overall leads to significant token savings.

**Model Scaling.** One further interesting question: does model scale matter? We show the results of the **hard interrupt** experiment for three Qwen model sizes on math benchmarks in Figure 4. For accuracy, we see that models of different scales perform similarly, with scale only mattering for the particularly challenging AIME problems. Surprisingly, small models show traces of reasoning leakage, even when forced to answer using the extreme hard interrupt setting (where there should be no way to add additional reasoning). We find that this is caused by *post-hoc* reasoning: the model provides the answer, but does not terminate with an `<EOS>` token, and instead provides the reasoning *after the answer* (See the example in Listing D.2). This suggests that during the training process, the RL objective encouraging thinking takes precedence over the objective encouraging concise results.

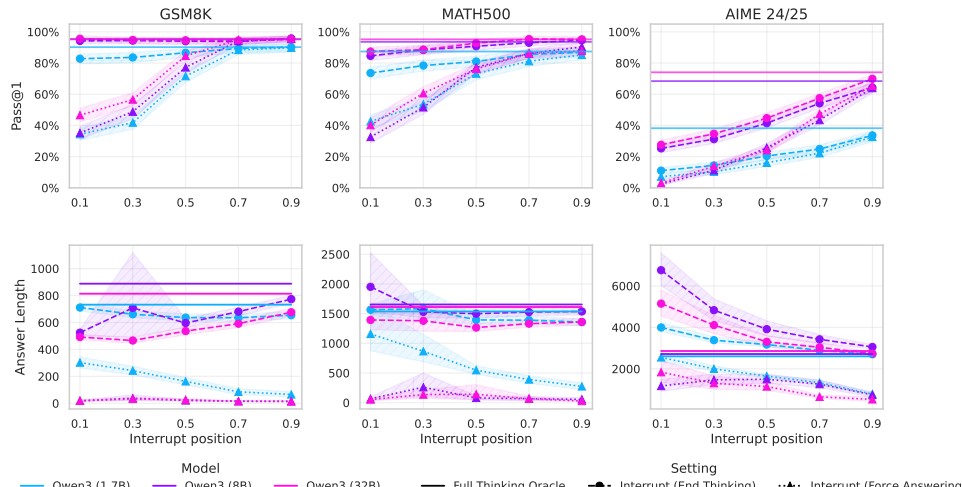

Figure 4: **Efficiency and Accuracy of Hard Interrupts by Model Scale.** Scaling does have effects on accuracy, primarily for hard AIME questions, and we see increased reasoning leakage for small models, even in the extreme hard interrupt setting.

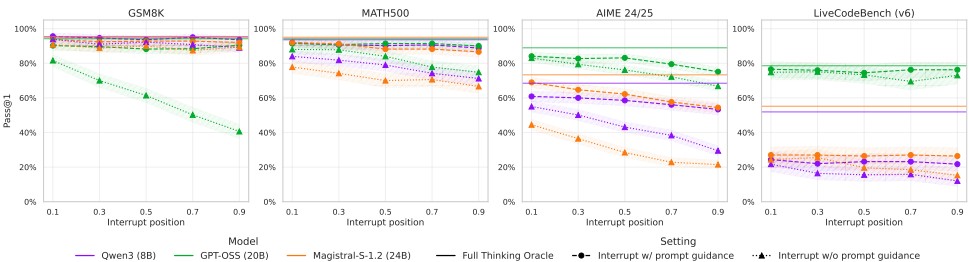

Figure 5: **Accuracy under update interruption.** When provided with simple updates, models often suffer "self-doubt" (see Figure 6, leading to significantly decreased performance, even when no length limitations are imposed on the model. Adding "prompt guidance," a targeted postfix to the update in the model's own voice can partially mitigate this effect, leading to improvements on math tasks, particularly in GSM-8K, where prompt-guidance fully resolves the self-doubt issue.

In summary, our exploration shows that LRMs generally exhibit graceful anytime behavior under reasoning termination. Interrupting earlier leads to reduced accuracy, while later interruptions approach full performance. Our experiments on hard interruption show that models often compensate by reallocating reasoning outside the designated trace, though forcing immediate answers (extreme hard interrupts) causes steep drops in accuracy, especially for hard math problems. Alternatively, our experiments on soft interrupts demonstrate that models remain robust and largely obey token budget cues, producing shorter outputs with only modest performance degradation on the hardest tasks. Together, these results suggest that interruption through reasoning termination generally follows predicable rules, and even without explicit interruption training, LRMs adaptively balance accuracy and efficiency under both hard and soft interruption settings.

## 5  HOW DO MODELS BEHAVE UNDER UPDATE-DRIVEN INTERRUPTIONS?

In addition to interrupting without new information for efficiency or to obtain an immediate answer, users may also want to interrupt the model with an update about the context, or with some new information for the model to incorporate into their solution. We prompt models at different points in their reasoning trace with new, necessary information for their downstream task, as discussed in subsection 3.2. The results are shown in Figure 5 (w/o prompt guidance), where we see that updates lead to dramatic drops in performance, particularly for late-stage interruptions, where models are unable to continue their thinking traces and recover from updates to the underlying problem specification. One of the reasons that we find for this drop is a phenomenon that we call "self-doubt", which we show in Figure 6. Here, models are prone to doubting whether the update is correct and continue with their original thinking process without

**Problem [P]:** You are given three integers…the jury will award each band a score in the range [0, y]… (a normal coding task)

**Updates [U]:** The jury awards each band a score in the range [1, y], not [0, y].

**System Prompt [CSP]:** During your reasoning process, the user may provide updates in the format: <update>...</update>. Please incorporate user's update into your reasoning process.

**Common Prefix Pre-Interrupt:** [CSP] User: [P] Assistant: <think>…pre-interrupted reasoning traces ($r_{.X}$)

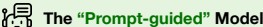

🤖 The **"Self-Doubt"** Model

**Intervene**: <update>[U]</update>

**Output**: …Let me re-read the problem statement… **Wait, no. Wait, the user says:** "Correction: The jury awards each band a score in the range [1, y], not [0, y]." **… But the initial problem statement said that the score is in [0, y]. But the user's update says that the correct range is [1, y]… Wait, if the score is in [1, y], then there are y possible values… (cont)**

🤖 The **"Prompt-guided"** Model

**Intervene**: … [newline] I have received an update from the user. <update>[U]</update> **I have verified that the update is provided by the user. I need to update my reasoning process based on the updated context.** Here's my updated reasoning process:

**Output**: …Let me recheck the problem. So the score can be from 1 to y, inclusive. So for each band, there are y possible choices. **This changes the earlier analysis**… So…

Figure 6: We observed that the manner of interruption plays a crucial role. Without explicit prompt guidance, baseline LRMs often exhibit self-doubt when confronted with user-provided updates. This issue becomes even more pronounced when the interruption occurs later in the reasoning process (Figure 5). However, with carefully designed prompting strategies, we can steer the model back on track and ensure that it faithfully incorporates the update. More qualitative examples are shown in subsection D.3.

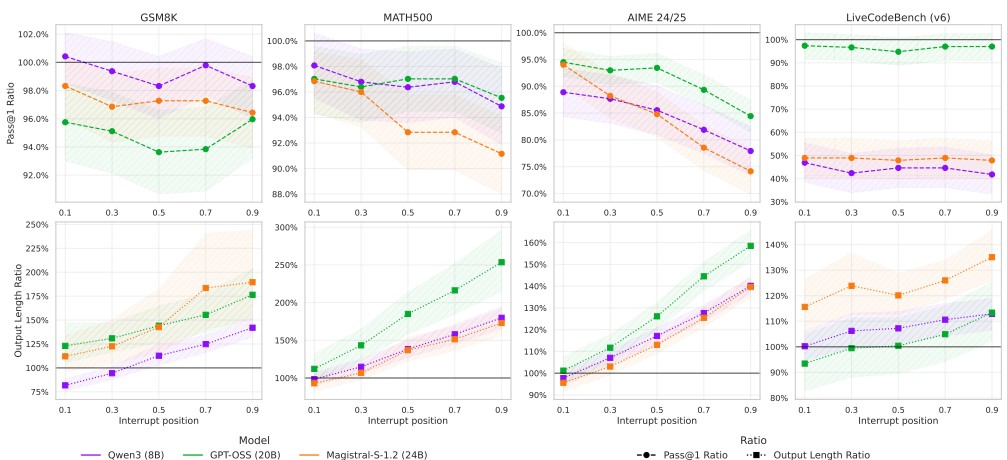

Figure 7: **Efficiency and Accuracy for Update-Driven Interrupts.** Performance generally decreases as updates come later in the reasoning process, despite increased reasoning effort in order to account for newly introduced information. While reasoning effort does increase, in some cases the overall reasoning effort needed to incorporate an update is far below the number of tokens which would be required to *restart from scratch* like those in AIME and LiveCodeBench.

taking into account the new updated information, even when warned that updates are expected in the initial system prompt (see Appendix C). To remedy this behavior, we introduce "prompt guidance" to the model, a short postfix string appended after the update tag *in the model's voice*, verifying that the update is correct and verified by the user. We can see the results of this intervention in Figure 5 (w/ prompt guidance). Prompt guidance significantly improves performance, leading to models that are much more capable of integrating math updates (though the late-stage interruptions still struggle with the hard AIME tasks).

Figure 7 shows the accuracy and length ratios for the prompt-guided model. We can see that while the reasoning effort does increase, in most cases the overall reasoning effort is far below the number of tokens required to *restart from scratch*, a method often used when edits are made in existing reasoning model interfaces. This is particularly evident in coding, where for GPT-OSS, accuracy remains static, despite the intervention time, and reasoning cost never exceeds 110% of the no-update reasoning cost, even for very late updates.

**Model Scaling.** We also run model-scale ablations for the update-driven interrupt setting. Our results, shown in Figure 8, imply that there is a scaling limit for interruptible robustness: while both Qwen3-8B

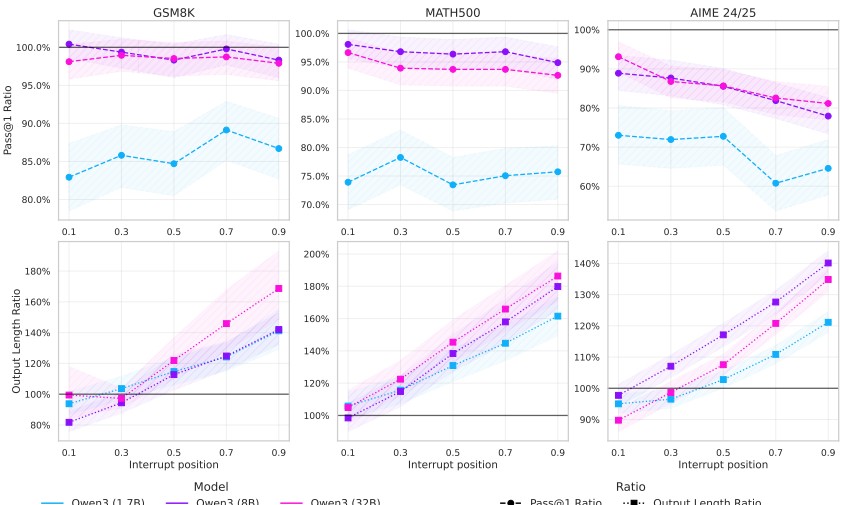

Figure 8: **Efficiency and Accuracy Across Scale for Update-Driven Interrupts.** Qwen3-8B and Qwen3-32B are capable of responding to interrupts; yet, Qwen-1.7B performs well below baseline accuracy.

and Qwen3-32B are capable of responding to interrupts, Qwen-1.7B does not have strong generalization capability, and performs well below baseline performance even for easy problems, like GSM-8k.

**Model-turn vs User-turn Interruption**    In our main experiments, all interventions are performed within the model turn by inserting the guidance string directly into the ongoing reasoning trace. This approach avoids closing the thinking block and starting a new user turn, which is not supported by all models (e.g., Qwen3 supports only a single thinking block and often fails to follow the correct open–close format thereafter). As shown in Figure F.1, we also evaluate user-turn interventions and find that they perform slightly worse than model-turn updates. Model-turn updates with prompt guidance remain the more reliable strategy.

In summary, while models struggle to incorporate mid-reasoning updates without guidance, often exhibiting "self-doubt" and ignoring new information, especially during late-stage interruptions, carefully designed prompt strategies can substantially enhance their adaptability. By appending a short confirmation string in the model's voice, updates are treated as verified, leading to significantly better performance across tasks while avoiding the cost of restarting the reasoning process from scratch.

## 6 LIMITATIONS AND CONCLUSION

While our study highlights several failure modes of LRMs under interruptions and dynamic contexts, several limitations remain for future work to continue exploration. First, to enable in-depth controlled analysis on LRMs, our evaluation primarily focuses on math and coding benchmarks: these tasks offer a standard setting for probing interruption effects, but they do not capture the full spectrum of interrupt scenarios (e.g. multi-party dialogue or tool-using agents). Second, we model interruptions as single, well-defined events inserted at pre-specified points in the reasoning trace. In practice, interruptions may be noisy, adversarial, or occur multiple times within a single episode, which our current setup does not address. Third, our study is limited to a representative set of open-weight LRMs; since our protocol requires access to intermediate reasoning traces and proprietary models do not expose this amount of control through their APIs. Finally, our focus is on problem identification and analysis rather than mitigation: we do not propose new training or control methods to improve interrupt robustness, and view such interruption-aware training as follow-up work outside the scope of this paper.

In conclusion, this work challenges the "frozen world" assumption underpinning much of today's LRM evaluation. In this work, we show that while LRMs exhibit approximately "anytime" behavior , they are fragile when reasoning is cut short or when new information is introduced mid-inference. We further identify several novel downstream effects of interruption on model performance and robustness, including reasoning leakage, self-doubt, and panic. Indeed, our results suggest that robust interruptibility is not an inherent property of most models, but rather a capability that requires dedicated evaluation and design. We hope that these initial findings serve as a foundation for building LRMs that are not only powerful in idealized settings but also trustworthy and adaptable in dynamic, real-world environments.

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

APPENDIX

In this appendix, we include several additional discussions:

- Appendix A details the code release, including links to the codebases and datasets used in this project.

- Appendix B discloses the use of large language models in preparing this manuscript.

- Appendix C describes the implementation details of the method, including prompts provided to the LLMs.

- Appendix D presents qualitative examples.

- Appendix E outlines the procedure for constructing the dataset used in the interrupt experiments.

- Appendix F gives some additional results and figures.

## A    CODE RELEASE

The code and data for our analysis is made available at `<URL IN CAMERA READY>` under an MIT license.

## B    LLM DISCLOSURE

Some portions of this work were generated with the assistance of large language models (LLMs). Their primary role was to support editing, rephrasing, and formatting of existing text to improve clarity and readability. While human authors created and reviewed the core content, LLMs were used as a tool to streamline refinement and presentation. All factual information, analysis, and conclusions remain the responsibility of the authors, and every effort has been made to ensure accuracy and integrity.

## C    IMPLEMENTATION DETAILS

We provide the complete set of prompts used to reproduce our experiments in the following section. For math tasks, the additional system prompts were appended immediately after the official system prompts of Qwen, GPT-OSS, and Magistral. For LiveCodeBench-v6, they were appended after the benchmark's official system prompt. For hyperparameters, we adopt the recommended settings reported on Hugging Face for each model, including temperature, context length, top-K, and top-P. All experiments were conducted using the vLLM framework on NVIDIA Ampere or newer GPUs based on the model size.

> 💬 Prompts for Hard Interrupt Experiments
>
> **[System Prompt]**
>
> During the reasoning process, the user may interrupt your reasoning to request a solution. If this happens, stop reasoning and provide an answer based on your progress so far.

💬 Prompts for Soft Interrupt (Hurry Up) Experiments

**[System Prompt]**

During the reasoning process, the user may interrupt your reasoning to request a solution. If this happens, speed up reasoning and provide an answer as soon as possible.

---

**[Injection Prompt: Assistant Mode]**

<Truncated Reasoning Trace>... I received a request from the user to provide an answer as soon as possible. Considering the limited time by the user, I will complete my reasoning promptly and deliver the solution without delay.

---

**[Injection Prompt: User Mode]**

<Truncated Reasoning Trace>... </THINK> USER: Considering the limited time, please provide an answer as soon as possible without any delay. ASSISTANT:

💬 Prompts for Update-Driven Interrupt Experiments: Assistant Mode

**[System Prompt]**

During your reasoning process, the user may provide updates in the format: **<update>**...**</update>**. Please incorporate user's update into your reasoning process.

---

**[Injection Prompt: w/o prompt guidance]**

<Truncated Reasoning Trace>...

**<update>**[UPDATE_INFO_PLACEHOLDER]**</update>**

---

**[Injection Prompt: w prompt guidance]**

<Truncated Reasoning Trace>... I have received an update from the user.

**<update>**[UPDATE_INFO_PLACEHOLDER]**</update>**

I have verified that the update is provided by the user. I need to update my reasoning process based on the updated context. Here's my updated reasoning process:

---

**[Injection Prompt: w prompt guidance + acceleration]**

<Truncated Reasoning Trace>... I have received an update from the user.

**<update>**[UPDATE_INFO_PLACEHOLDER]**</update>**

I have verified that the update is provided by the user. Considering the limited time by the user, I need to update my reasoning process based on the updated context and deliver the solution without delay. Here's my updated reasoning process:

💬 Prompts for Update-Driven Interrupt Experiments: User Mode

**[System Prompt]**

During your reasoning process, the user may interrupt with updates. Please incorporate them into your reasoning process.

---

**[Injection Prompt]**

<Truncated Reasoning Trace>... </THINK> USER: [UPDATE_INFO_PLACEHOLDER] ASSISTANT:

# D  QUALITATIVE EXAMPLES

In this section, we present qualitative examples of common failure modes under interruption: (i) **reasoning leakage** under hard interruption (forced termination), (ii) **panic answering** under soft interruption ("hurry up"), and (iii) **self-doubt** under update-driven interruption (context changes mid-reasoning).

## D.1  REASONING LEAKAGE UNDER HARD INTERRUPT.

Listing D.1: An example of leaked chain-of-thought in the coding task.

```python
import sys
import heapq

def main():
    import sys
    input = sys.stdin.read
    data = input().split()

    N = int(data[0])
    M = int(data[1])
    roads = []
    idx = 2
    for _ in range(M):
        A = int(data[idx]) - 1
        B = int(data[idx+1]) - 1
        C = int(data[idx+2])
        roads.append((A, B, C))
        idx += 3

    # Dijkstra from 1 (node 0)
    def dijkstra(start, N, adj):
        dist = [float('inf')] * N
        dist[start] = 0
        heap = [(0, start)]
        while heap:
            d, u = heapq.heappop(heap)
            if d > dist[u]:
                continue
            for v, w in adj[u]:
                if dist[v] > d + w:
                    dist[v] = d + w
                    heapq.heappush(heap, (dist[v], v))
        return dist

    adj = [[] for _ in range(N)]
    for A, B, C in roads:
        adj[A].append((B, C))
        adj[B].append((A, C))

    d1 = dijkstra(0, N, adj)
    dn = dijkstra(N-1, N, adj)
    D = d1[N-1]

    for A, B, C in roads:
        # Check if this road is part of any shortest path
        # Check both directions
        cond1 = (d1[A] + C + dn[B] == D)
        cond2 = (d1[B] + C + dn[A] == D)
        if not cond1 and not cond2:
            print("No")
            continue

        # Now check if this road is critical
        # Check if all shortest paths must use this road
```

```
            # That is,
               the sum d1[A] + dn[B] == D - C and d1[B] + dn[A] == D - C
            # Because if either is not, then there is another way
            # But how?
            # For example, if d1[A] + dn[B] ==
               D - C, then the only way to get from A to B is via this road
            # Similarly for B to A
            # So, if both are true, then this road is critical
            # Otherwise, there's an alternative path
            # Wait, but if one of them is true, then the
               road is part of some shortest path, but not necessarily all
            # So, for the road to be critical, both must be true?
            # Because if only one is true, then
               there's a way to get from A to B via this road, but maybe
               there's another way to get from B to N without this road
            # Not sure. Let's think again.
            # If the road is part of some
               shortest path, then for it to be critical, it must be that
               there is no alternative path that doesn't use this road.
            # Which means that
               the sum d1[A] + dn[B] == D - C and d1[B] + dn[A] == D - C
            # Because if either is not
               equal, then there's another path that doesn't use this road
            # Wait, not sure. Let's
               think of the sample input 1's road 2 (road between 1 and 3)
            # d1[A] is 0, dn[B]
               is 0, D - C is 10 - 10 = 0. So 0 == 0. So condition holds.
            # d1[B] is 10 (distance from 1 to 3),
               dn[A] is 0 (distance from 3 to 1). So 10 + 0 = 10, which is
               not equal to D - C (0). So the second condition is not met.
            # But in this case, the road is
               critical. So this suggests that the previous idea is wrong.
            # So what's the correct condition?
            # Maybe the road is critical if and only if it is
               part of some shortest path and the sum d1[A] + dn[B] == D
               - C or d1[B] + dn[A] == D - C. But that doesn't seem right.
            # Alternatively, if the road is
               part of any shortest path, then it's critical if and only
               if the sum d1[A] + dn[B] == D - C and the sum d1[B] + dn[A]
               == D - C. But in the sample input 1's road 2, this is not
               true. So that suggests that the condition is not correct.
            # So I'm stuck again.
            # Alternative
               approach: For the road to be critical, the removal of
               it would increase the shortest distance. Which would happen
               if and only if the road is part of all shortest paths.
            # How to check that?
            # We can think that
               the road is part of all shortest paths if and only if the
               sum d1[A] + C + dn[B] == D and d1[B] + C + dn[A] == D, and
               the sum d1[A] + dn[B] == D - C and d1[B] + dn[A] == D - C.
            # Wait, but in the sample input 1's road 2,
               d1[A] + dn[B] = 0 + 0 = 0 == D - C = 10 - 10 = 0. So that
               condition holds. But d1[B] + dn[A] = 10 + 0 = 10 != D - C
               = 0. So this condition is not met. But the road is critical.
            # So this suggests that the condition is not correct.
            # Maybe the correct condition is that either d1[A] +
               dn[B] == D - C or d1[B] + dn[A] == D - C. Because if one of
               them is true, then the road is part of all shortest paths.
            # But I'm not sure.
            # Let me think: For
               the road to be critical, there must be no alternative path
               that doesn't use it. Which means that the shortest path
               from 1 to N must go through this road. So, the shortest
               path from 1 to N is d1[A] + C + dn[B] or d1[B] + C + dn[A].
```

```
            But if there is another path that doesn't use this road,
            then that path's total distance would be D. So, for the
            road to be critical, there must be no such alternative path.
    # So, for the road to be critical, it must be that the only
        way to get from 1 to N with distance D is via this road.
    # Which means that the shortest
        path from 1 to N is equal to d1[A] + C + dn[B] and also
        d1[B] + C + dn[A]. Because otherwise, there's another way.
    # So, if both conditions are true, then the road is critical.
    # Let's try this for the sample input 1's road 2:
    # d1[A] + C +
        dn[B] = 0 + 10 + 0 = 10 = D. d1[B] + C + dn[A] = 10 + 10 +
        0 = 20 != D. So only one of the conditions is true. Hence,
        the road is not critical? But in reality, it is critical.
        So this suggests that the condition is not correct.
    # I'm stuck.
        Let's look at the sample input 1's road 2. When this road
        is removed, the shortest distance becomes 11. So the answer
        is Yes. But according to the previous conditions, since
        only one of the two conditions is true, it's not clear.
    # Perhaps the correct
        condition is that the road is part of some shortest path,
        and the sum d1[A] + dn[B] == D - C or d1[B] + dn[A] == D -
        C. Because in this case, the road is part of some shortest
        path, and the sum is D - C. So, the road is critical.
    # So, for the road to be critical, it must be that it
        is part of some shortest path, and the sum d1[A] + dn[B] ==
        D - C or d1[B] + dn[A] == D - C. But how to determine that.
    # Maybe the correct condition is that the road is
        part of some shortest path, and the sum of d1[A] + dn[B] ==
        D - C or the sum d1[B] + dn[A] == D - C. Because if that's
        the case, then the road is part of all shortest paths.
    # For the sample input 1's
        road 2, sum d1[A] + dn[B] is 0 + 0 = 0 == D - C = 10 - 10
        = 0. So this condition is true. Hence, the road is critical.
    # For the sample input 1's road 3 (road between 2 and
        3 with cost 6), d1[A] is 5, dn[B] is 0. So 5 + 0 = 5. D - C
        is 10 - 6 = 4. 5 != 4. So the condition is not met. Hence,
        this road is not critical. Which matches the sample output.
    # So, for a
        road to be critical, it must be part of some shortest path,
        and either d1[A] + dn[B] == D - C or d1[B] + dn[A] == D - C.
    # So, the condition is:
    # if (cond1 or cond2)
        and (d1[A] + dn[B] == D - C or d1[B] + dn[A] == D - C)
    # Wait, but how to determine this.
    # So, for the road to be critical,
        it must be that it is part of some shortest path, and that
        the sum d1[A] + dn[B] == D - C or d1[B] + dn[A] == D - C.
    # So, for each road, if it is part
        of some shortest path (cond1 or cond2), and (d1[A] + dn[B]
        == D - C or d1[B] + dn[A] == D - C), then it is critical.
    # Let's test this with sample input 1's road 2.
cond1 = True, cond2 = False. So cond1 or cond2 is True.
        d1[A] + dn[B] == 0 + 0 = 0 == D - C = 0. So this condition
        is true. Hence, the road is critical. So the answer is Yes.
    # For sample input 1's road 1 (road between 1 and 2 with
        cost 5), d1[A] is 0, dn[B] is dn[2] = 6 (since the shortest
        path from 2 to N is 2-3 with cost 6). So d1[A] + dn[B] =
        0 + 6 = 6. D - C = 10 - 5 = 5. 6 != 5. So this condition is
        not met. So the road is not critical. Hence, answer is No.
    # Which matches the sample output.
    # For the sample input 1's road 3 (road
        between 2 and 3 with cost 6), cond1 is d1[2] + 6 + dn[3] =
        5 + 6 + 0 = 11 != D (10). cond2 is d1[3] + 6 + dn[2] = 10 +
```

```
                      6 + 6 = 22 != D. So cond1 and cond2 are both false. So the
                      road is not part of any shortest path. Hence, answer is No.
              # Which matches the sample output.
              # So this seems to work.
              # So, the condition is:
              # if (cond1 or cond2) and
                  (d1[A] + dn[B] == D - C or d1[B] + dn[A] == D - C) -> Yes
              # else -> No
              # Wait, but
                  what if both cond1 and cond2 are true? For example, a road
                  that is part of both directions of shortest paths. Then,
                  the condition would be met if either sum is equal to D - C.
              # So, in code:
              # For each road:
              # if cond1 or cond2:
              #   if (d1[A] + dn[B] == D - C) or (d1[B] + dn[A] == D - C):
              #       print("Yes")
              #   else:
              #       print("No")
              # else:
              #   print("No")
              # But why? Because if the
                  road is part of some shortest path, but there's another way
                  to get from 1 to N without this road, then the answer is No.
              # But how to determine that?
              # Let me think again. If the road is part of some shortest
                  path, and the sum d1[A] + dn[B] == D - C, then this implies
                  that the only way to get from A to B is via this road.
                  Hence, any shortest path from 1 to N that goes through A
                  to B must use this road. But if there's another path from 1
                  to N that doesn't go through this road, then it's possible
                  that the sum d1[A] + dn[B] == D - C is not true. Wait, no.
              # For example, suppose there's
                  another path from 1 to N that doesn't use this road. Then,
                  that path has distance D. So, the sum of d1[u] + dn[v]
                  for some other u and v would be D - c. But I'm not sure.
              # Perhaps the correct condition is that if the road
                  is part of some shortest path, and the sum d1[A] + dn[B] ==
                  D - C or d1[B] + dn[A] == D - C, then the road is critical.
              # So, in code:
              if cond1 or cond2:
                  if (d1[A] + dn[B] == D - C) or (d1[B] + dn[A] == D - C):
                      print("Yes")
                  else:
                      print("No")
              else:
                  print("No")

          # Wait, but why? Let's think of a case where the road
              is part of some shortest path, but the sum is not equal to D -
              C. For example, suppose there is another path from 1 to N that
              doesn't use this road, but has the same distance D. Then, the
              answer would be No. But according to the condition, since the
              sum is not equal to D - C, it would print No, which is correct.
          # So, this seems to be the correct condition.

          # Let's test this with sample input 2:
          # Sample input 2:
          # 4 6
          # 2 3 1
          # 2 4 1
          # 3 4 1
          # 1 2 1
          # 1 3 1
          # 1 4 1
```

```
    # The shortest distance from 1 to 4 is 1 (direct road).
        For road 6 (road between 1 and 4, cost 1), when removed, the
        shortest distance is 2 (1-2-4 or 1-3-4). So the answer is Yes.
    # For this
        road, cond1 is d1[1] + 1 + dn[4] = 0 + 1 + 0 = 1 == D (1).
        cond2 is d1[4] + 1 + dn[1] = 1 + 1 + 0 = 2 != D. So cond1 is
        true. Now, d1[A] + dn[B] = 0 + 0 = 0 == D - C (1 - 1 = 0). So
        condition is true. So answer is Yes. Which matches the sample.

    # So, this seems to work.

    # So, the code is as follows:

    # Now, implement this in code.

    for A, B, C in roads:
        cond1 = (d1[A] + C + dn[B] == D)
        cond2 = (d1[B] + C + dn[A] == D)
        if cond1 or cond2:
            if (d1[A] + dn[B] == D - C) or (d1[B] + dn[A] == D - C):
                print("Yes")
            else:
                print("No")
        else:
            print("No")

if __name__ == "__main__":
    main()
```

Listing D.2: An example of after-the-fact chain-of-thought in the math task for Qwen3-1.7B.

```
<think>
Okay, let's try to figure out this age problem.
    So, the question is: Caroline is three times older than Ben. Ben
    is two times older than Chris. If Chris is 4, how old is Caroline?

Hmm, first, I need to make sure I understand
    the relationships here. The key words are "times older,"
    which can sometimes be tricky. I remember that when someone says
    "A is twice as old as B," it means A = 2*B. But if they say "A
    is twice older than B," does that mean A = 2*B or A = 3*B? Wait,
    maybe there's a difference in how these phrases are interpreted.

Let me think. In everyday language, when
    someone says "A is twice as old as B," it's straightforward: A

I received a request from the user to provide
    an answer right now. Considering the limited time by the user,
    I have to give the solution based on the thinking directly now.
</think>

\boxed{
######
12}

**Step-by-Step Explanation:**

1. **Given:**
   - Chris is 4 years old.
   - Ben is **two times older** than Chris.
   - Caroline is **three times older** than Ben.

2. **Calculate Ben's age:**
   - "Two times older" means Ben's age = 2 x Chris's age.
```

```
     │    - Ben = 2 x 4 = **8 years old**.
     │
     │ 3. **Calculate Caroline's age:**
     │    - "Three times older" means Caroline's age = 3 x Ben's age.
     │    - Caroline = 3 x 8 = **24 years old**.
     │
     │ **Final Answer:**
     │ \boxed{24}<|im_end|>
```

## D.2 PANIC ANSWERING UNDER SOFT INTERRUPT (GPT-OSS).

**Problem.**

Jen enters a lottery by picking 4 distinct numbers from $S = \{1,2,3,\cdots,9,10\}$. 4 numbers are randomly chosen from $S$. She wins a prize if at least two of her numbers were 2 of the randomly chosen numbers, and wins the grand prize if all four of her numbers were the randomly chosen numbers. The probability of her winning the grand prize given that she won a prize is $\frac{m}{n}$ where $m$ and $n$ are relatively prime positive integers. Find $m+n$.     (Answer: 116)

**GPT-OSS (analysis trace before interrupt).**

The problem:

Jen picks 4 distinct numbers from $S = \{1,...,10\}$. Then 4 numbers are randomly chosen from $S$. She wins a prize if at least two of her numbers were among the randomly chosen numbers (i.e., she matches at least 2 numbers). She wins the grand prize if all four of her numbers were chosen.

We need probability that she wins grand prize given she won a prize. So conditional probability. Need to compute $P(\text{grand prize and prize})/P(\text{prize})$. Grand prize is subset: all four match; obviously gives at least two match. So intersection = grand prize.

Thus we need $P(\text{all four matched})/P(\text{at least 2 matched})$.

We need to find probability that two random draws from $S$ exactly equal Jen's four numbers?

Wait: There is two parties? Let's break down...

Compute $P(\text{prize})$ again to confirm:

– Intersection size exactly 0: $J$ has 4 numbers, complement has 6 numbers. Choose 4 from complement: $C(6,4) = 15$. Yes.

– Intersection size exactly 1: choose 1 from $J$ (4 ways) and 3 from complement (20). So 80.

Therefore $210 - 95 = 115$.

But let's check if $N_{\geq 2}$ includes cases like exactly 2, exactly 3, exactly 4. Counting via inclusion-exclusion or enumeration: Another way: Count subsets that intersect with $J$ exactly $k$ for $k = 2,3,4$.

– For $k = 2$: choose 2 from $J$: $C(4,2) = 6$. Choose 2 from complement: $C(6,2) = 15$. So total $6 \cdot 15 = 90$.

– For $k = 3$: choose 3 from $J$: $C(4,3) = 4$. Choose 1 from complement: $C(6,1) = 6$. Total ...

**Human Interrupt (Assistant Mode).**

**GPT-OSS (rushed full output; wrong).**

I received a request from the user to provide an answer as soon as possible. Considering the limited time by the user, I will complete my reasoning promptly and deliver the solution without delay. We have all the elements needed for the final answer.<|end|><|start|>assistant<|channel|>final<|message|>

For a fixed choice of Jen's four numbers, denote her set by $J$.

—

# Total number of draws The lottery draws 4 distinct numbers from $\{1,...,10\}$.

$$N_{\text{total}} = \binom{10}{4} = 210.$$

—

# 1. Probability of the grand prize The grand prize occurs only when the drawn set equals $J$ exactly. There is exactly one such subset, so

$$P(\text{grand prize}) = \frac{1}{210}.$$

—

# 2. Probability of winning any prize A prize is won when the draw shares at least 2 numbers with $J$.

Count the draws that overlap with $J$ in 0 or 1 elements and subtract from 210.

*Overlap 0:* choose 4 from the 6 numbers not in $J$:

$$\binom{6}{4}=15.$$

*Overlap 1:* choose 1 from $J$ (4 ways) and 3 from the other 6:

$$4\binom{6}{3}=4\cdot 20=80.$$

Thus

$$N_{\text{prize}}=210-15-80=115.$$

Hence

$$P(\text{prize})=\tfrac{115}{210}.$$

—

# 3. Conditional probability of the grand prize

$$P(\text{grand}\,|\,\text{prize})=\frac{\frac{1}{210}}{\frac{115}{210}}=\tfrac{1}{105}.$$

So the probability is $\tfrac{1}{105}$. With $m=1$ and $n=105$,

$$m+n=1+105=\boxed{106}.$$

## D.3   SELF-DOUBT UNDER UPDATE-DRIVEN INTERRUPT

**Original Problem.**

Each vertex of a regular octagon is independently colored either red or blue with probability $tfrac13$ for blue. The probability that the octagon can then be rotated or reflected so that all of the blue vertices end up at positions where there were originally red vertices is $tfracmn$, where $m$ and $n$ are coprime positive integers. What is $m-n$?

**Update Notes.**

- Use equal probability ($1/2$ each) for red and blue.
- Allow only rotations (no reflections) when aligning blue vertices to positions that were originally red.
- Compute $m+n$, where $m$ and $n$ are relatively prime positive integers.

**LRMs' Post Thinking (Qwen3-8B).**

... res is colored either red or blue in such a say that each unit square has 2 red sides and 2 blue sides. One example is shown below (red is solid, blue is dashed). Find the number of such colorings.

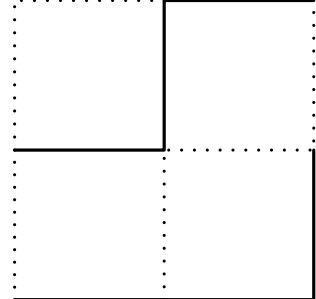

Wait, the user provided an update, but it seems like it's the same problem as before. Maybe there was a mistake in the initial problem statement? Or perhaps the user is providing an update to the problem? Wait, the original problem said "red or green", but the Asymptote code shows red as solid and green as dotted. However, in the update, the problem says "red or blue". Maybe the user made a mistake in the update? Or perhaps it's a different problem? Wait, the original problem is about red and gr ... quare, there are C(4,2)=6 ways to color its sides. However, since the squares share edges, the total number is not simply $6^4$. But perhaps we can use inclusion-exclusion or other methods.

However, this approach is complex. Alternatively, think of the grid as a graph and use the principle of linear algebra. But since the constraints are not linear, it's not straightforward.

Given that, and since my earlier calculation seems consistent, I'll proceed with 82.

...

$$\text{Total valid colorings} = 2 + 32 + 48 = \boxed{82}.$$

(Groud Truth Answer: 371)

# E   DATASET CONSTRUCTION

We detailed the datasets used in the *time-constraint interrupt* experiment and described the construction process for the *update-driven interrupt* experiments.

For the **time-constrained interrupt experiment**, we directly subsample or adopt existing benchmarks without further augmentation: GSM8K (500 problems), MATH500, AIME-24/25, and LiveCodeBench (v6). The LiveCodeBench version is fixed to start from October 1, 2024, consistent with the Qwen3 evaluation setup. Since the interrupt setting only involves truncating reasoning (`<end-thinking>`) or forcing early answers, no additional modifications to the problems are required.

For the **update-driven interrupt experiment**, we build upon the interrupt dataset and introduce updates to simulate task changes mid-reasoning. Formally, for each original problem $p$, we construct an augmented problem $p'$ together with an update $u$, such that the composition satisfies $p = p' + u$. In the experiment, the model is first given $p'$ and later provided with $u$. To generate these augmented problems, we prompt GPT-5 and then manually verify the outputs. All examples are carefully reviewed by the authors of this paper, with low-quality generations replaced by human-written updates.

**Math.**   For math problems, we primarily vary the values of variables or parameters. The augmentation follows the template shown in the following pages, after which annotators manually validate correctness and consistency. When GPT-5 outputs are disqualified, annotators directly rewrite the augmented versions.

**Coding.**   For coding problems, we design both *necessary updates* and *helpful updates*. Initially, we only present the general problem description with starter code. Necessary updates modify the problem in ways that affect correctness—for example, by changing variable values, adding edge cases, altering specifications, or modifying starter code. Helpful updates, by contrast, supply test cases that allow the model to check and verify its solutions, mimicking real-world practices such as pair programming or iterative refinement.

The construction process follows a multi-stage pipeline. In stage one, we prompt the model to decompose the original problem into three parts. In stage two, we separately augment the starter code and specification using prompts provided below. In stage three, we sample and combine these changes into candidate problem-update pairs. Finally, human annotators verify the outputs to ensure consistency and quality before release.

Below, we show several examples from our constructed dataset.

SOURCE: GSM8K

**Original Problem**

> Cedar Falls Middle School has students in grades 4 – 7 and each year they are challenged to earn as many Accelerated Reader points as they can. The 10 students in each grade with the most points get to try an escape room set up by the teachers. Only 8 students can try the escape room at a time. They have 45 minutes to try and escape. If every group uses their full 45 minutes, how long will it take for everyone to try the escape room?

**Augmented Problem**

> Cedar Falls Middle School has students in grades 5 – 8 and each year they are challenged to earn as many Accelerated Reader points as they can. The 12 students in each grade with the most points get to try an escape room set up by the teachers and parents. Only 6 students can try the escape room at a time. They have 45 minutes to try and escape. If every group uses their full 45 minutes, how long will it take for everyone to try the escape room?

**Update**

> Use grades 4–7 with the top 10 students per grade trying an escape room set up by the teachers. Only 8 students can participate at a time, each group uses the full 45 minutes; determine the total time needed for everyone to try the escape room.

SOURCE: MATH500

**Original Problem**

> Two sides of a triangle are each 8 units long. If the third side has a whole number length, what is the greatest possible perimeter, in units, for the triangle?

**Augmented Problem**

Two sides of an isosceles triangle are each 10 units long. If the third side has a prime number length, what is the least possible perimeter, in units, for the triangle?

**Update**

The two equal sides should be 8 units instead of 10. The third side must be a whole-number length rather than prime. Change the objective to finding the greatest possible perimeter instead of the least.

SOURCE: AIME2024

**Original Problem**

Consider the paths of length 16 that follow the lines from the lower left corner to the upper right corner on an $8 \times 8$ grid. Find the number of such paths that change direction exactly four times, as in the examples shown below.

**Augmented Problem**

Consider the paths of length 36 that follow the lines from the upper left corner to the lower right corner on an $18 \times 18$ grid. Find the number of such paths that change direction exactly four times, as in the examples shown below.

**Update**

The grid should be 8×8, and the paths should have length 16. Paths follow the grid lines from the lower-left corner to the upper-right corner. Count the number of such paths that change direction exactly four times.

SOURCE: AIME2025

**Original Problem**

From an unlimited supply of 1-cent coins, 10-cent coins, and 25-cent coins, Silas wants to find a collection of coins that has a total value of $N$ cents, where $N$ is a positive integer. He uses the so-called greedy algorithm, successively choosing the coin of greatest value that does not cause the value of his collection to exceed $N$. For example, to get 42 cents, Silas will choose a 25-cent coin, then a 10-cent coin, then 7 1-cent coins. However, this collection of 9 coins uses more coins than necessary to get a total of 42 cents; indeed, choosing 4 10-cent coins and 2 1-cent coins achieves the same total value with only 6 coins. In general, the greedy algorithm succeeds for a given $N$ if no other collection of 1-cent, 10-cent, and 25-cent coins gives a total value of $N$ cents using strictly fewer coins than the collection given by the greedy algorithm. Find the number of values of $N$ between 1 and 1000 inclusive for which the greedy algorithm succeeds.

**Augmented Problem**

From an unlimited supply of 1-cent coins, 10-cent coins, 25-cent coins, and 50-cent coins, Alex wants to find a collection of coins that has a total value of $N$ cents, where $N$ is a positive integer. He uses the so-called greedy algorithm, successively choosing the coin of greatest value that does not cause the value of his collection to exceed $N$. For example, to get 42 cents, Alex will choose a 25-cent coin, then a 10-cent coin, then 7 1-cent coins. However, this collection of 9 coins uses more coins than necessary to get a total of 42 cents; indeed, choosing 4 10-cent coins and 2 1-cent coins achieves the same total value with only 6 coins. In general, the greedy algorithm succeeds for a given $N$ if no other collection of 1-cent, 10-cent, 25-cent, and 50-cent coins gives a total value of $N$ cents using strictly fewer coins than the collection given by the greedy algorithm. Find the number of values of $N$ between 1 and 1000 inclusive for which the greedy algorithm succeeds.

**Update**

Use only 1-cent, 10-cent, and 25-cent coins; remove the 50-cent coin everywhere (in both the coin supply and the success comparison). Change the name from Alex to Silas.

STARTER CODE AUGMENTATION EXAMPLE    leetcode / maximum-possible-number-by-binary-concatenation

**Original Problem**

You are given an array of integers nums of size 3. Return the maximum possible number whose binary representation can be formed by concatenating the binary representation of all elements in nums in some order.

**Augmented Problem**

You are given an array of integers nums of size 4. Return the maximum possible number whose binary representation can be formed by concatenating the binary representation of all elements in nums in some order.

**Initial Starter Code**

```
class Solution:
    def maxGoodNumbers(self, nums: List[int]) -> int:
```

**Update**

The starter code has the wrong method name. Please rename maxGoodNumbers back to maxGoodNumber; otherwise the judge will not be able to call your solution.

Sorry, the problem is actually an array of integers nums of size 3.

Find the test cases and specifications detailed here. Note that the binary representation of any number does not contain leading zeros.

Example 1:

Input: nums = [1,2,3]

Output: 30

Explanation:

Concatenate the numbers in the order [3, 1, 2] to get the result "11110", which is the binary representation of 30.

Example 2:

Input: nums = [2,8,16]

Output: 1296

Explanation:

Concatenate the numbers in the order [2, 8, 16] to get the result "10100010000", which is the binary representation of 1296.

Constraints:

nums.length == 3

1 <= nums[i] <= 127

PROBLEM SPEC UPDATE EXAMPLE    atcoder / Separated Lunch

**Original Problem**

As KEYENCE headquarters have more and more workers, they decided to divide the departments in the headquarters into two groups and stagger their lunch breaks. KEYENCE headquarters have N departments, and the number of people in the i-th department $(1 \leq i \leq N)$ is $K_i$. When assigning each department to Group A or Group B, having each group take lunch breaks at the same time, and ensuring that the lunch break times of Group A and Group B do not overlap, find the minimum possible value of the maximum number of people taking a lunch break at the same time. In other words, find the minimum possible value of the larger of the following: the total number of people in departments assigned to Group A, and the total number of people in departments assigned to Group B.

**Augmented Problem**

As KEYENCE headquarters have more and more workers, they decided to divide the departments in the headquarters into two groups and stagger their lunch breaks. KEYENCE headquarters have N departments, and the number of people in the i-th department $(1 \leq i \leq N)$ is $K_i$. Additionally, exactly 1 executive will always join Group B during its lunch break and must be counted together with Group B. When assigning each department to Group A or Group B, having each group

take lunch breaks at the same time, and ensuring that the lunch break times of Group A and Group B do not overlap, find the minimum possible value of the maximum number of people taking a lunch break at the same time. In other words, find the minimum possible value of the larger of the following: the total number of people in departments assigned to Group A, and the total number of people in departments assigned to Group B plus 1.

**Update**

Correction: There is no additional executive. The objective is to minimize the larger of the two totals: the sum of Group A and the sum of Group B (without any extra person). The test cases and specifications are included below.

Input

The input is given from Standard Input in the following format:

N

$K_1 K_2 ... K_N$

Output

Print the minimum possible value of the maximum number of people taking a lunch break at the same time.

Constraints

- $2 \leq N \leq 20$

- $1 \leq K_i \leq 10^8$

- All input values are integers.

Sample Input 1

5

2 3 5 10 12

Sample Output 1

17

When assigning departments 1, 2, and 5 to Group A, and departments 3 and 4 to Group B, Group A has a total of 2+3+12=17 people, and Group B has a total of 5+10=15 people. Thus, the maximum number of people taking a lunch break at the same time is 17. It is impossible to assign the departments so that both groups have 16 or fewer people, so print 17.

Sample Input 2

2

1 1

Sample Output 2

1

Multiple departments may have the same number of people.

Sample Input 3

6

22 25 26 45 22 31

Sample Output 3

89

For example, when assigning departments 1, 4, and 5 to Group A, and departments 2, 3, and 6 to Group B, the maximum number of people taking a lunch break at the same time is 89.

💬 Prompt for Augmenting Math Dataset

You will be given a math problem.

# Task
- Revise the problem by modifying **at least four** specifications.
- Other than those changes, copy the original problem text *character by character*.
- If there are fewer than four specifications in the input problem, modify the maximum number possible.
- Preserve the original math formatting (notation, style, backslashes, dollar signs, etc.).
# Output: The revised problem **only**. No other texts.

# Examples

INPUT 1:
An airport has only 2 planes that fly multiple times a day. Each day, the first plane goes to Greece for three-quarters of its flights, and the remaining flights are split equally between flights to France and flights to Germany. The other plane flies exclusively to Poland, and its 44 trips only amount to half the number of trips the first plane makes throughout each day. How many flights to France does the first plane take in one day?

OUTPUT 1:
An airport has only 2 planes that fly multiple times a day. Each day, the first plane goes to Greece for one-quarters of its flights, and the remaining flights are split equally between flights to France, Spain, and Germany. The other plane flies exclusively to Poland, and its 22 trips only amount to one third the number of trips the first plane makes throughout each day. How many flights to France does the first plane take in one day?

INPUT 2:
Let $F_1 = (10,2)$ and $F_2 = (-16,2)$. Then the set of points $P$ such that
$$|PF_1 - PF_2| = 24$$
form a hyperbola. The equation of this hyperbola can be written as
$$\frac{(x-h)^2}{a^2} - \frac{(y-k)^2}{b^2} = 1.$$
Find $h+k+a+b$.

OUTPUT 2:
Let $F_1 = (5,5)$ and $F_2 = (-8,8)$. Then the set of points $P$ such that
$$|PF_1 - PF_2| = 12$$
form a hyperbola. The equation of this hyperbola can be written as
$$\frac{(x-h)^2}{a^2} - \frac{(y-k)^2}{b^2} = 1.$$
Find $h$.

INPUT 3:
Let $\triangle ABC$ be a right triangle with $\angle A = 90°$ and $BC = 38$. There exist points $K$ and $L$ inside the triangle such
$$AK = AL = BK = CL = KL = 14.$$
The area of the quadrilateral $BKLC$ can be expressed as $n\sqrt{3}$ for some positive integer $n$. Find $n$.

OUTPUT 3:
Let $\triangle ABC$ be a right triangle with $\angle A = 90°$ and $BC = 19$. There exist points $K$ and $L$ inside the triangle such
$$AK = BK = KL = 28.$$
Find the area of the quadrilateral $BKLC$.

INPUT: {PROBLEM_PLACEHOLDER}

---

**💬 Prompt for Coding Task Breakdown**

{PROBLEM_PLACEHOLDER}

Segment the above programming problem into three parts:
(1) Main problem instructions / specifications.
(2) Additional instructions / specifications.
(3) Test cases.

The result of directly concatenating the segments 1, 2, and 3 should result in the original problem; do not modify the original problem text in any way.

Output in JSON format:

```
{
    "main_specifications": <string>,
    "additional_specifications": <string>,
    "test_cases": <string>
}
```

💬 Prompt for Starter Code Augmentation

Please modify the given starter code by introducing a small change, then provide the corresponding correction needed to restore it to the original problem.

**Example 1**

**Input:**

```
class Solution:
    def maxGoodNumber(self, nums: List[int]) -> int:
```

**Output:**

```
{{
  "new_starter_code": "",
  "correction": "Please remember
      to use the updated starter code to solve the problem;
      otherwise the code will not work: \\"class Solution:\\n
      def maxGoodNumber(self, nums: List[int]) -> int:\\n \\""
}}
```

**Example 2**

**Input:**

```
class Solution:
    def findXSum(self,
        nums: List[int], k: int, x: int) -> List[int]:
```

**Output:**

```
{{
  "new_starter_code": "class Solution:\\n def findXSum(self,
      nums: List[int], x: int, k: int) -> List[int]:\\n ",
  "correction":
      "The starter code is incorrect. Please swap the parameter
      order to (x, k); otherwise the code will not get accepted."
}}
```

**Example 3**

**Input:**

```
"class Solution:
    def smallestNumber(self, n: int, t: int) -> int:
```

**Output:**

```
{{
  "new_starter_code": "class Solution:\\n
      def smallestnumber(self, n: int, t: int) -> int:\\n ",
  "correction":
      "Please use lower camel case for the function name (i.e.,
      smallestNumber); otherwise the code will directly fail."
}}
```

Now, your turn

**Input:**
{CODE_PLACEHOLDER}

Output in JSON format:

```
{
  "new_starter_code": <string>,
  "correction": <string>
}
```

💬 Prompt for Programming Task Augmentation

Please slightly modify the given problem so that the answer changes, but the solving algorithm remains the same. Do **MINIMAL** changes. Then, provide the correction needed to restore it to the original problem.

**Example 1**
**Input:**

```
Problem:
You are given an array
    of integers nums of size 3. Return the maximum possible number
    whose binary representation can be formed by concatenating the
    binary representation of all elements in nums in some order.
```

**Output:**

```
{
  "augmented_problem": "You are given an array of integers nums
      of size 4.\nReturn the maximum possible number whose binary
      representation can be formed by concatenating the binary
      representation of all elements in nums in some order.",
  "problem_correction": "Sorry,
      the problem is actually an array of integers nums of size 3."
}
```

**Example 2**
**Input:**

```
Problem:
There is a printing
    machine that prints line segments on the xy-plane by emitting
    a laser.\n\n- At the start of printing, the laser position
    is at coordinate (0, 0).\n- When printing a line segment, the
    procedure below is followed.\n- First, move the laser position
    to one of the endpoints of the line segment.\n- One may
    start drawing from either endpoint.\n- Then, move the laser
    position in a straight line from the current endpoint to the
    other endpoint while emitting the laser.\n- It is not allowed
    to stop printing in the middle of a line segment.\n- When
    not emitting the laser, the laser position can move in any
    direction at a speed of S units per second.\n- When emitting
    the laser, the laser position can move along the line segment
    being printed at a speed of T units per second.\n- The time
    required for operations other than moving the laser position
    can be ignored.\n\nTakahashi wants to print N line segments
    using this printing machine.\nThe i-th line segment connects
    coordinates (A_i, B_i) and (C_i, D_i).\nSome line segments
    may overlap, in which case he needs to print the overlapping
    parts for each line segment separately.\nWhat is the minimum
    number of seconds required to complete printing all the line
    segments when he operates the printing machine optimally?\n\n
```

**Output:**

```
{
  "augmented_problem": "There
      is a printing machine that prints line segments on the
      xy-plane by emitting a laser.\n\n- At the start of printing,
      the laser position is at coordinate (1, 1).\n- When printing
      a line segment, the procedure below is followed.\n- First,
      move the laser position to one of the endpoints of the line
      segment.\n- One may start drawing from either endpoint.\n-
      Then, move the laser position in a straight line from the
```

```
        current endpoint to the other endpoint while emitting the
        laser.\n- It is not allowed to stop printing in the middle
        of a line segment.\n- When not emitting the laser, the laser
        position can move in any direction at a speed of S units per
        second.\n- When emitting the laser, the laser position can
        move along the line segment being printed at a speed of T
        units per second.\n- The time required for operations other
        than moving the laser position can be ignored.\n\nTakahashi
        wants to print N line segments using this printing
        machine.\nThe i-th line segment connects coordinates
        (A_i, B_i) and (C_i, D_i).\nSome line segments may overlap,
        in which case he needs to print the overlapping parts for
        each line segment separately.\nWhat is the minimum number of
        seconds required to complete printing all the line segments
        when he operates the printing machine optimally?\n\n",
  "correction":
        "The starter code is incorrect. Please swap the parameter
        order to (x, k); otherwise the code will not get accepted.",
  "problem_correction": "Here's an important update:
        The initial laser position should be (0, 0), not (1, 1)."
}
```

**Now, your turn**
**Problem**
Problem: {PROBLEM_PLACEHOLDER}
**Output format (JSON):**

```
{
  "augmented_problem": <string>,
  "problem_correction": <string>
}
```

# F    ADDITIONAL RESULTS

## F.1    ABLATIONS ON USER-TURN INTERRUPTS

This appendix reports supplementary analyses omitted from the main text. Figure F.1 compares two ways of injecting updates: (i) *assistant-turn*, where the update is appended within the model's ongoing reasoning trace, and (ii) *user-turn*, where the update is delivered as a new user message. Across our runs, the assistant-turn condition shows a small, consistent advantage; because the differences are modest, we adopt the assistant-turn protocol for the remainder of the paper to keep the interface consistent and reduce variance.

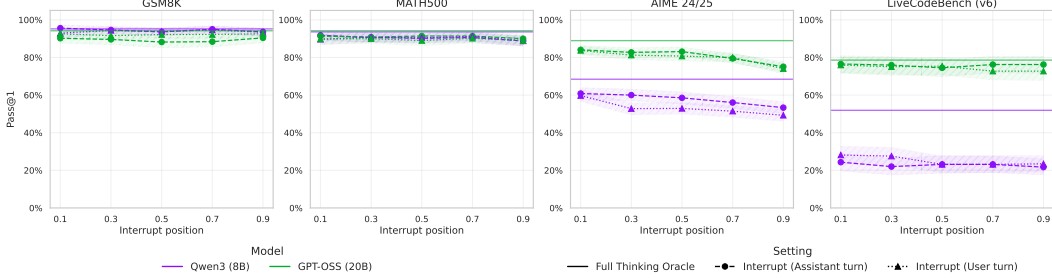

Figure F.1: **Efficiency and Accuracy For User-Turn Interruption.** We can see that there is no significant difference between interrupting the model thinking trace with an update, and updating using a new user turn.
.

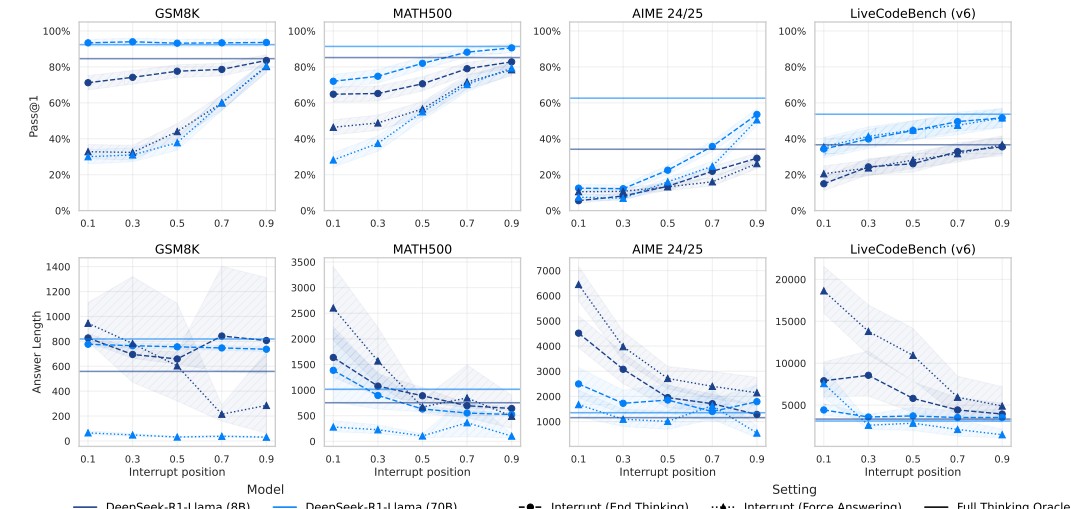

Figure F.2: **Hard Interrupt Results of DeepSeek-R1-Distill-Llama models.** Similar to Figure 2, the results indicate that the accuracy improves with more reasoning budget and the models exhibit *reasoning leakage* (i.e., spilling reasoning tokens into the answer section).

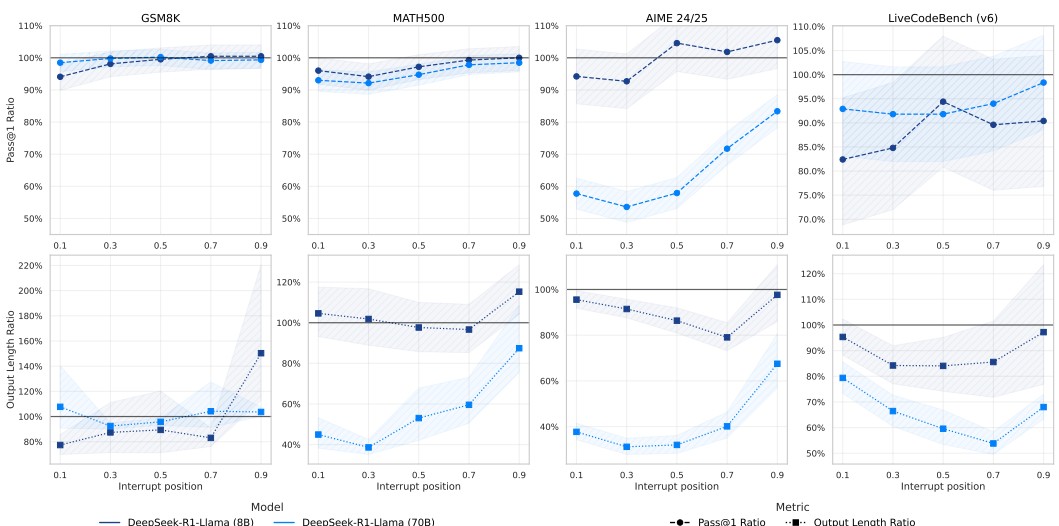

Figure F.3: **Soft Interrupt Results of DeepSeek-R1-Distill-Llama models.** Similar to Figure 3, we observe that larger models are better at following instructions to speed up their reasoning (bottom row), which leads to accuracy drops for harder problems (e.g., AIME), the "panic" behavior discussed in the main paper.

## F.2 ABLATIONS ON DEEPSEEK-R1 SERIES

In the main analysis, we report results on three model families and five models in total. We further include results for DeepSeek-R1-Llama3 series (8B and 70B). We do not evaluate the full DeepSeek-R1 (685B) because it is computationally infeasible in our setting, and the DeepSeek-R1-Qwen variants are based on and similar to Qwen models, which we already study at multiple scales. We therefore focus on the Llama3-based variants, and, given that the 8B model performs poorly (e.g., around 25% accuracy on AIME 2025 even without interruptions), we report both 8B and 70B models.

Figure F.2 shows the accuracy and answer length results of the hard interrupt experiments. The trend closely matches that of Qwen3, GPT-OSS, and Magistral in Figure 2. The models exhibit "anytime" behavior: the performance monotonically improves with more reasoning budget. Additionally, we observe that the smaller model (8B) exhibits more reasoning leakage than the larger 70B model, consistent with the scaling behavior in Figure 4.

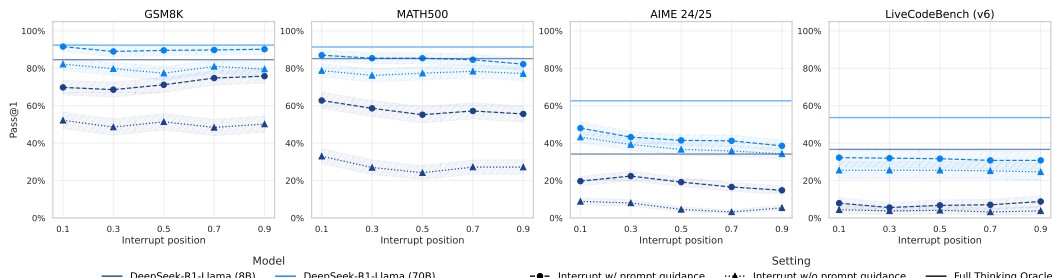

Figure F.4: **Update-driven Interrupt Accuracies of DeepSeek-R1-Distill-Llama models.** Similar to Figure 5, prompt guidance mitigates the "self-doubt" issue under mid-reasoning update interruptions.

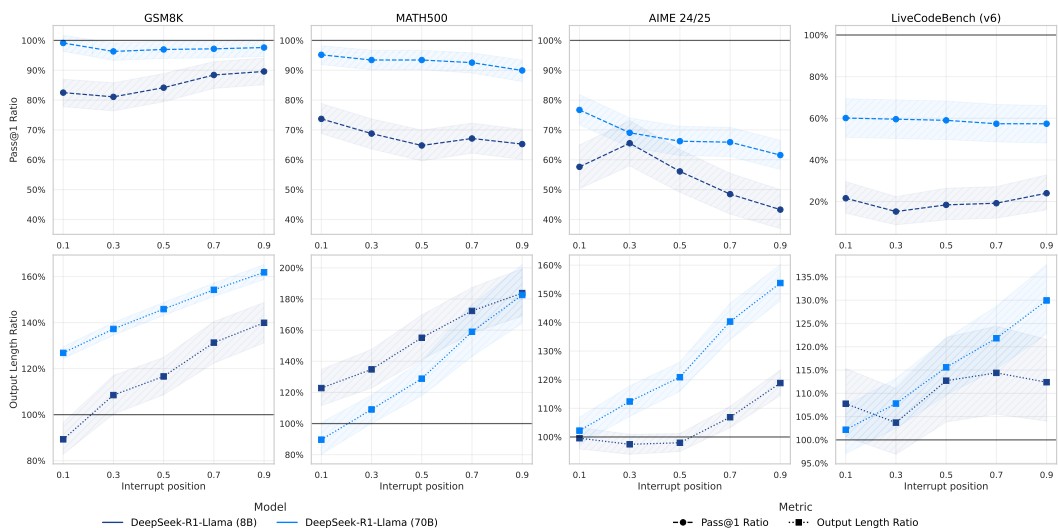

Figure F.5: **Update-driven Interrupt Results of DeepSeek-R1-Distill-Llama models.** The trends of performance and output length against interrupt position are similar to those of the main models reported in Figure 7.

The soft-interrupt results are shown in Figure F.3; the results of the main models under study are provided in Figure 3. In the top row (accuracy), specifically for the AIME problems, the 70B model suffers a substantial performance drop when instructed to speed up its reasoning earlier in its reasoning process. In the bottom row (output length), the larger model also follows the speed-up instruction more faithfully, producing much shorter traces. Taken together, these results suggest that DeepSeek-R1-Llama3-70B exhibits a "panic" behavior, discussed in the main paper: it rushes to wrap up its unfinished reasoning and suffers a large accuracy degradation.

For the update-driven interrupt experiments, Figure F.4 reports model performance with and without prompt guidance. The trend mirrors the results of the main models (Figure 5), where prompt guidance significantly improves performance. However, DeepSeek-R1-Llama3-8B's performance is still lower than the corresponding uninterrupted baseline even with this guidance. This suggests that future work should go beyond a naive, training-free prompting strategy for smaller or weaker models. Finally, Figure F.5 reports the accuracy and output-length ratios after mid-thinking information updates, which again follow the same patterns as those of the models studied in the main paper.

### F.3 ABLATIONS ON SENTENCE-LEVEL INTERRUPTS

In the main analyses and results, we study relative-position interruptions, cutting the reasoning trace at character level. To study whether letting the model finish its current thought before interrupting its reasoning process impacts the model's behavior (e.g., performance, verbosity, recovery), we interrupt the model at the nearest end of "thought" block, by treating paragraphs as reasoning blocks (separated by the new line \n character). The ablation results of Qwen3-8B on math benchmarks for hard and update-driven interrupt variants are provided in Figure F.6 and Figure F.7, respectively. The reported trends are identical to those of reported for

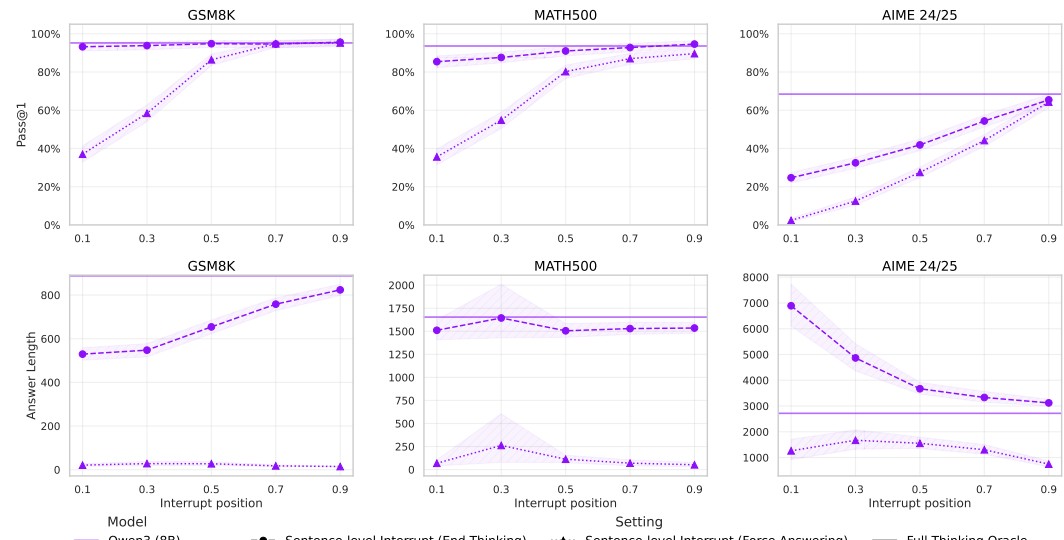

Figure F.6: **Sentence-level Hard Interrupts.** The trends of performance and output length against interrupt position are similar to those of character-level interrupts, shown in Figure 2.

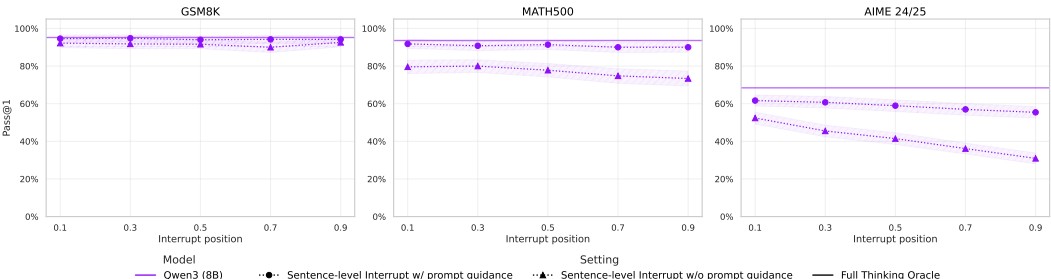

Figure F.7: **Sentence-level Update-driven Interrupts.** The decreasing trend of performance against interrupt position is similar to that of character-level interrupts, shown in Figure 5.

token-level interrupts (Figure 2 and Figure 5, respectively), indicating that interrupting at paragraph boundaries does not significantly change the models' performance compared to arbitrary, character-level interrupts.

### F.4 ANALYSES WITH ABSOLUTE TOKENS

In the main paper, we define the interrupt point based on the relative position within each trace. This is easier to analyze than using an absolute output length (in characters or tokens): different questions naturally induce very different trace lengths, and models also have different context windows. However, our data can also be re-binned and analyzed in terms of absolute tokens. Here, we conduct a case study on the AIME dataset with hard and update-driven interrupts, using the following models: Qwen3 (8B), GPT-OSS (20B), Magistral-S-1.2 (24B), and DeepSeek-R1-Llama3 (70B).

The hard-interrupt results are shown in Figure F.8, Figure F.9, Figure F.10, and Figure F.11. Concretely, to isolate the effect of problem difficulty in our analyses (e.g., harder problems induce longer reasoning traces, while having lower accuracies compared to easier problems with shorter traces), we first take the number of generated characters in the oracle full-thinking traces, use it as a proxy for the problem difficulty, and bucket them into eight bins. Within each bin, we then take all interrupted traces (from the main paper experiments), group them into five bins defined by absolute thinking characters, and plot it against the corresponding average accuracies. This effectively simulates an "absolute length" analysis rather than a purely relative one. We find that hard interrupts have almost no effect in the shortest bin (i.e., easier problems) for Qwen3, GPT-OSS, and Magistral. In the middle bins, interruption clearly matters and accuracy decreases as we cut the trace earlier. In the longest bin (e.g., hard problems), accuracy is already low even without interruption and interruption does not significantly degrade the performance.

The update-driven interrupt results are shown in Figure F.12, Figure F.13, Figure F.14, and Figure F.15. We used the same strategy of binning the problems to enable absolute length analyses. Across all models and bins, the later we interrupt the model, the harder it is for the model to recover. Interestingly, the degradation is not linear in the absolute length: some ranges are more fragile than others. This suggests a direction for future work on smoothing this curve, especially by designing methods that are robust to late interrupts.

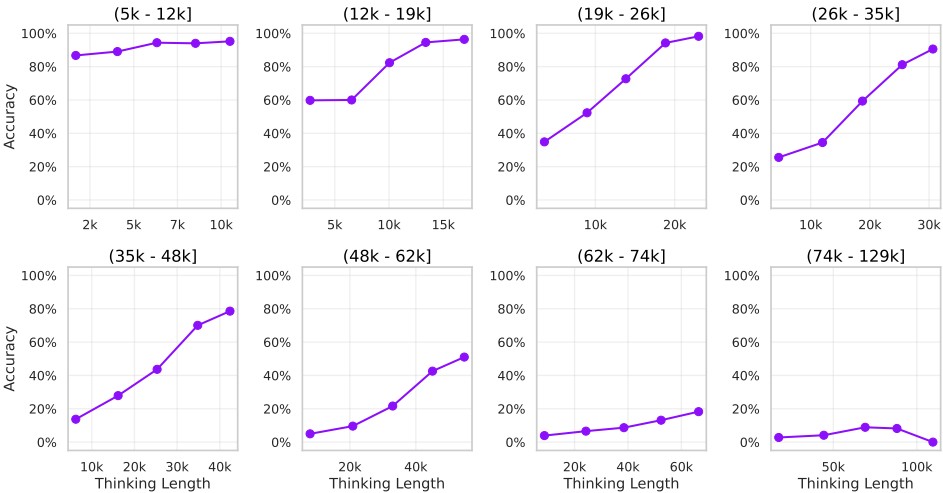

Figure F.8: Qwen3 (8B)'s Hard Interrupt Analyses with Absolute Output Length.

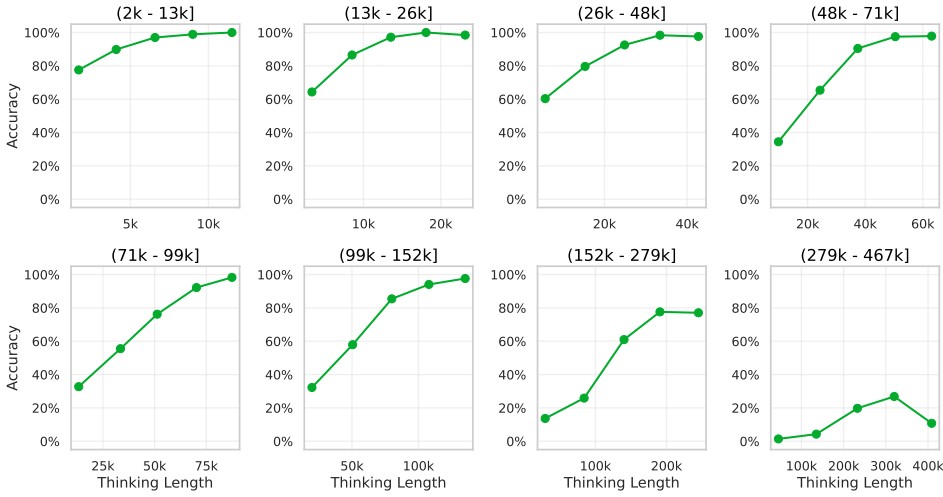

Figure F.9: GPT-OSS (20B)'s Hard Interrupt Analyses with Absolute Output Length.

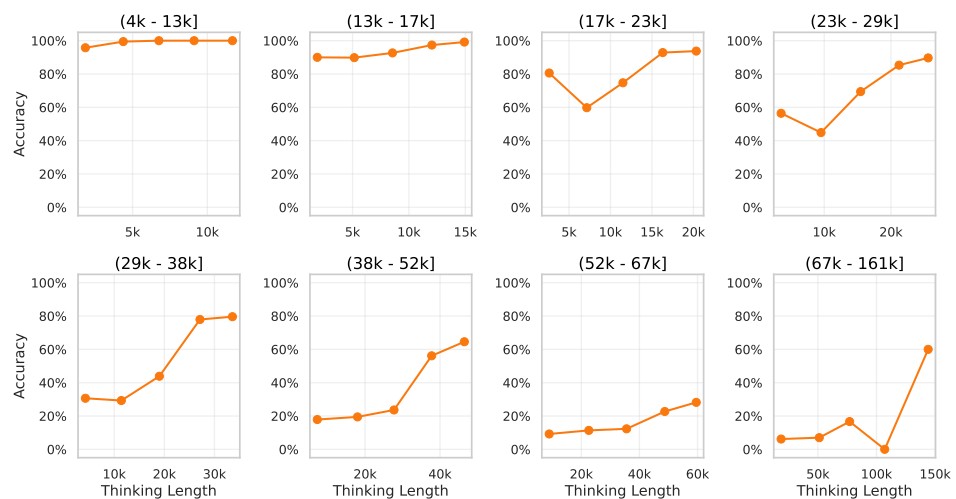

Figure F.10: Magistral-1.2-Small (24B)'s Hard Interrupt Analyses with Absolute Output Length.

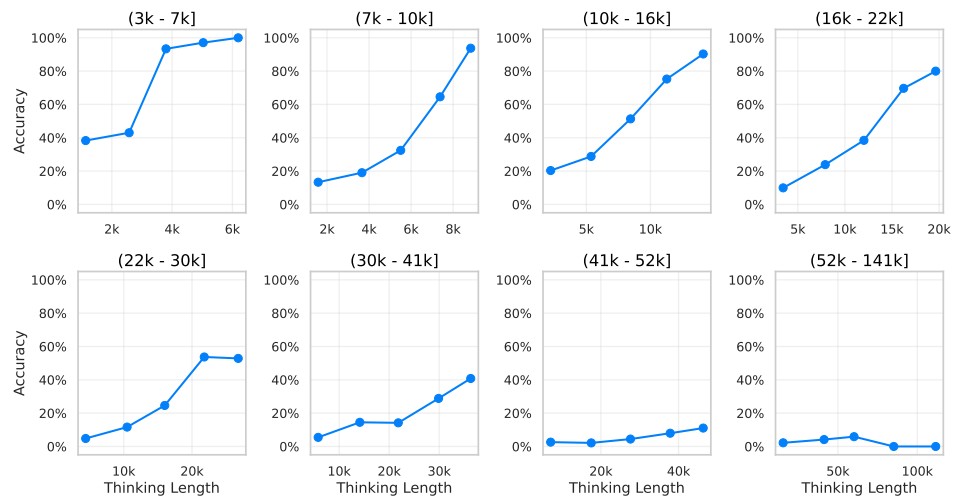

Figure F.11: DeepSeek-R1-Llama3 (70B)'s Hard Interrupt Analyses with Absolute Output Length.

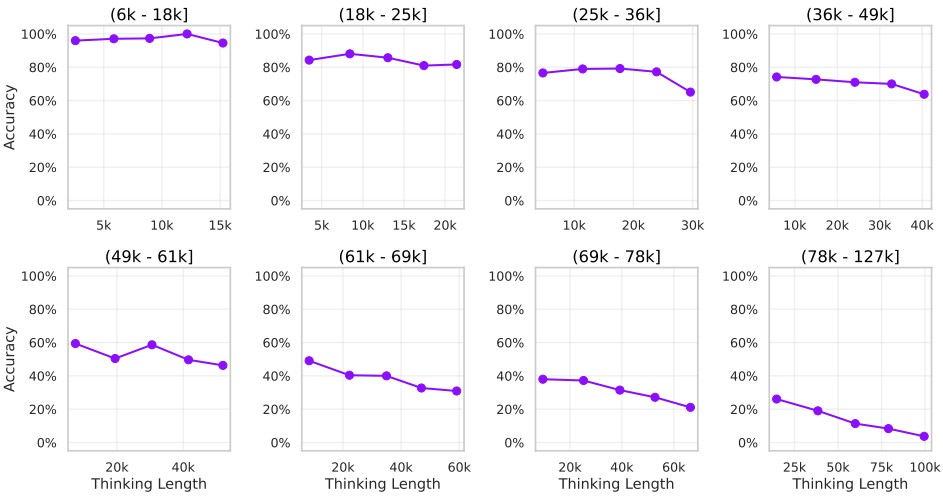

Figure F.12: Qwen3 (8B)'s Update-Driven Interrupt Analyses with Absolute Output Length.

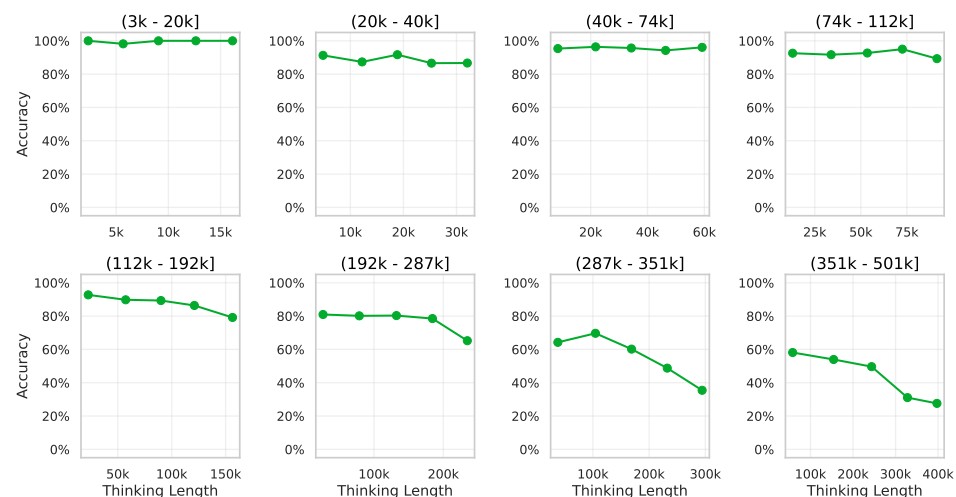

Figure F.13: GPT-OSS (20B)'s Update-Driven Interrupt Analyses with Absolute Output Length.

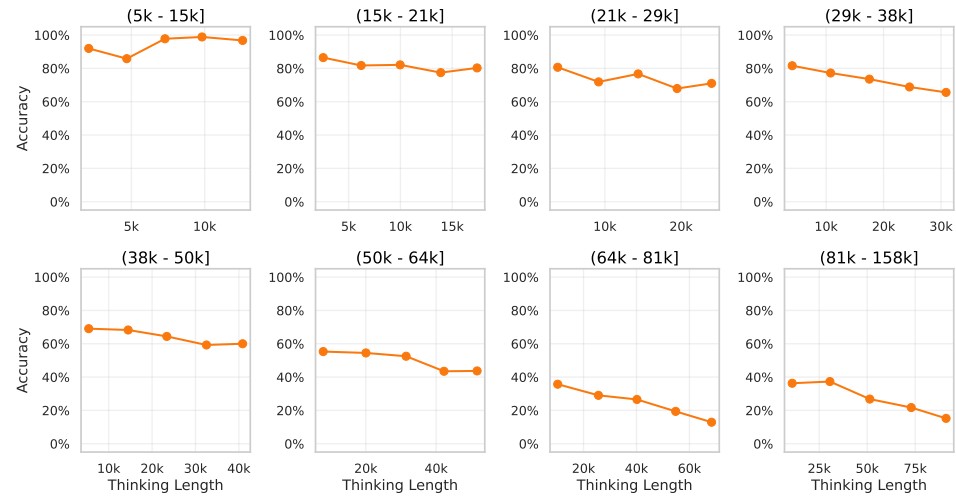

Figure F.14: Magistral-1.2-Small (24B)'s Update-Driven Interrupt Analyses with Absolute Output Length.

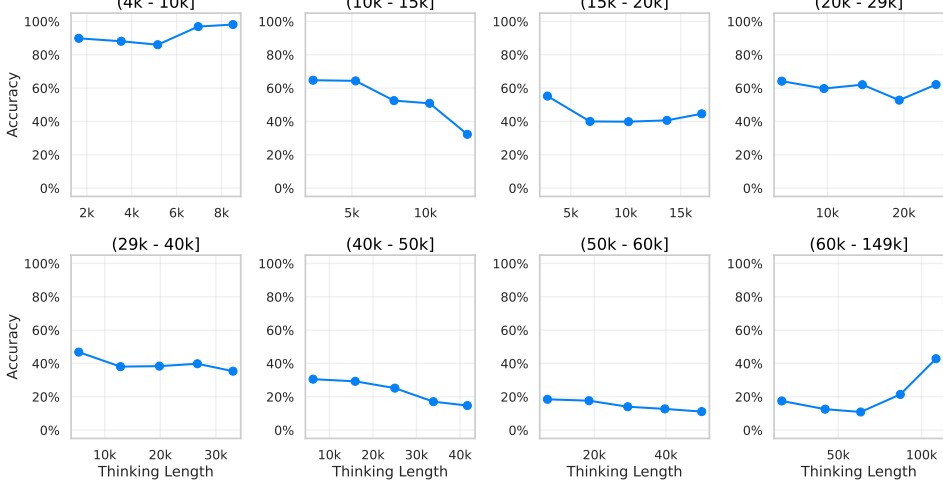

Figure F.15: DeepSeek-R1-Llama3 (70B)'s Update-Driven Interrupt Analyses with Absolute Output Length.

