# OpenReview forum: "Are Large Reasoning Models Interruptible?"
_ICLR.cc/2026/Conference — Submitted to ICLR 2026_

### Official Review · Reviewer_w9LU · 2025-10-28

**Soundness:** 3
**Presentation:** 3
**Contribution:** 3
**Rating:** 6
**Confidence:** 4

**Summary:**

This paper challenges the “frozen world” assumption underlying most reasoning-model evaluations—the idea that model reasoning unfolds in static, unchanging contexts. It systematically investigates how Large Reasoning Models (LRMs) behave when their reasoning process is interrupted mid-inference or when the context dynamically changes. Two main scenarios are explored: (1) Time-constrained interruptions, where the model is prompted to stop reasoning early (“hard interrupts”) or to accelerate reasoning (“soft interrupts”); and (2) Update-driven interruptions, where new information is injected mid-reasoning, requiring the model to adapt to revised problem specifications. The authors design dedicated evaluation protocols across mathematical and coding benchmarks, analyze distinctive failure modes such as reasoning leakage, self-doubt, and panic, and propose a simple prompt-guidance technique to improve update incorporation.

**Strengths:**

- The paper addresses a genuine gap by moving beyond static evaluations and introducing interruptibility as a new dimension of reasoning robustness. This perspective is practically relevant for real-world applications such as interactive agents and long-running inference systems.

- The proposed evaluation protocol systematically covers both time-constrained and update-driven interruptions, yielding useful insights across multiple datasets and models. The experimental design is clear.

- The identification of reasoning leakage, self-doubt, and panic offers interpretable and human-intuitive failure categories that help conceptualize how reasoning breaks down under dynamic conditions.

- The proposed prompt-guidance method is easy to implement, and empirically effective in improving model stability and update incorporation—without requiring additional training or model modification.

**Weaknesses:**

- While the paper exposes the vulnerability of reasoning models to interruptions, it does not propose solution for this issue. The introduced prompt-guidance strategy is heuristic and only partially mitigates the issue, offering limited insight into how truly interruption-resilient reasoning architectures might be designed.
- Experiments are limited to open-source LRMs (e.g., Qwen3, GPT-OSS, Magistral). Validation on closed systems such as Gemini 2.5 Pro or GPT-5 would strengthen the claim that interruptibility reflects a general property of reasoning models.
- Although the paper presents interruptibility as a novel dimension of reasoning robustness, similar mechanisms have been explored in prior studies on token-budget control and inference-time adaptation. The conceptual distinction, especially between soft interrupts and thinking budget, should be articulated more clearly.
- The use of relative fractions of the reasoning trace implicitly depends on model verbosity and generation length. This may bias the interpretation of “early” versus “late” interruptions across models with differing reasoning styles.

**Questions:**

Please see the weaknesses.

---

> ### Author Response · Authors · 2025-11-19
> **Rebuttal by Authors (1)**
>
> Thanks for highlighting the contributions of our work, including the novel problem setup, thorough evaluation, and useful insights about failure modes of LRMs under dynamic settings. We respond to the reviewer’s concerns below:
>
> ---
>
> **[W1] Problem setup and evaluation vs Solution**
>
> The primary goal of the paper is to (1) establish a foundational, controlled evaluation protocol and analysis of interruptions in LRMs and (2) to clearly identify and characterize the associated failure modes. The main contribution is therefore problem setup / identification and systematic evaluation / analysis, rather than proposing a fully engineered solution.
>
> We agree with the reviewer that prompt guidance alone is unlikely to be an definitive fix to the identified problem. We intentionally frame our prompt-guidance method as a lightweight intervention that can be applied across existing models to probe how much behavior can be improved without retraining. We clarify this positioning and contribution more explicitly in the revised Limitations section and highlight interruption-aware training as a promising direction for future work.
>
> ---
>
> **[W2] Closed-source model evaluation**
>
> We agree that including frontier closed-source systems would further strengthen our empirical results and insights. However, our interruption protocol requires mid-thinking control over the model’s internal reasoning stream (e.g., truncating or updating the ongoing chain of thought at specified token positions). Current APIs for systems such as GPT-5 and Gemini-2.5-Pro expose only turn-level interfaces: the user provides a prompt, the model returns a completed answer, and there is no supported mechanism to cut off or edit the intermediate reasoning trace. Because of that, we cannot run interrupted inference on closed-source reasoning models. We have mentioned this constraint in the updated Limitations section.
>
> ---
>
> **[W3] Prior work on token-budget control and inference-time adaptation**
>
> We appreciate the reviewer’s pointer to token-budget and adaptive inference methods.
>
> As discussed in the original related work (Lines 116-136), we view these as complementary but conceptually distinct methods. Specifically, these methods assume pre-defined budget and constraints (e.g., a fixed token budget or latency target) and design inference strategies to stay within the specified limits. In contrast, consistent with real-world user and context interruption scenarios, we explicitly study dynamic interruptions, where the model / system is **not aware in advance** when and how its reasoning will be cut off or its context updated. In addition to hard and soft interruptions, we also study update-driven interruptions, consistent with our motivation of testing model robustness under dynamic environment interruptions.
>
> Our experiments therefore focus on the dynamics of mid-trajectory interruption: how accuracy, stability, and failure modes evolve as a function of where in the reasoning trace the interruption occurs, rather than on minimizing total tokens or runtime. In the updated paper, we already clarify this distinction more explicitly and highlight that our framework could in principle be combined with token-budget control methods; for example, one can design interruption-aware, adaptive policies that anticipate potential cuts rather than assuming a fixed budget.

---

> > ### Author Response · Authors · 2025-11-19
> > **Rebuttal by Authors (2)**
> >
> > **[W4] Relative-position interruption**
> >
> > To address the reviewer’s concern about relative-position interrupts, we used our existing data to analyze the models’ performance on AIME problems against the number of absolute thinking tokens. Concretely, to isolate the effect of problem difficulty in our analyses (e.g., harder problems induce longer reasoning traces, while having lower accuracies compared to easier problems with shorter traces), we first take the number of generated characters in the oracle full-thinking traces, use it as a proxy for the problem difficulty, and bucket them into eight bins. Within each bin, we then take all interrupted traces (from the main paper experiments), group them into five bins defined by absolute thinking characters, and plot it against the corresponding average accuracies. This effectively simulates an “absolute length” analysis
> > rather than a purely relative one. We run the above analysis on hard interrupt (i.e., end thinking) and update-driven interrupt settings. We find trends consistent with our initial results based on relative-position interrupts: (1) performance improves when interrupted later (Figures F.8-F.11), and (2) performance declines when the updates are provided later (Figures F.12-F.15). This additional view leads to consistent qualitative conclusions, while the relative-fraction parameterization remains more interpretable for cross-model comparison and for characterizing interruption dynamics.
> >
> > We would also like to highlight that our objective is not to rank models by raw speed or efficiency, but to understand how each model’s behavior changes under interruption relative to its own uninterrupted baseline. Problem difficulty and typical context/trace length vary both across benchmarks and models, so a fixed absolute number of thinking tokens does not correspond to a comparable “stage” of reasoning across settings. Using relative positions (e.g., interrupting at 10%, 20%, etc. of the model’s own thinking trace) allows us to align experiments to comparable phases of the model’s reasoning process, making pathologies directly comparable across models and tasks.

---

### Official Review · Reviewer_2ARY · 2025-10-31

**Soundness:** 3
**Presentation:** 3
**Contribution:** 3
**Rating:** 6
**Confidence:** 4

**Summary:**

This paper introduces a new evaluation suite that is designed specifically to measure the robustness of reasoning models when interrupt during reasoning, as well as context changes during reasoning.
The experments are done using math reasoning and programming tasks.
The results show that current reasoning models can often fail to handle interruption or context changes during reasoning.
The paper shows the effect of introducing interruption at different reasoning lengths and demonstrates results that can be meaningful for future model development.

**Strengths:**

1. The problem this paper aims to show is an important one for actual model deployment.
2. The investigation of this paper covers many aspect of the proposed challenge of interruption that would be a useful resource.

**Weaknesses:**

1. The evaluations in this paper, while informative, still focuses on tasks that could be completed in a short period of time. Therefore, I think the proposed issue of interruption and context change may not happen if the reasoning time is relevatively short.
2. The paper could be stronger if it includes more models to compare with, such as Deepseek etc.
3. The evaluations are mostly shown in a plot, it would be more clearer if the paper could introduce a single number metric to describe the behavior of the models when interrupt.

**Questions:**

1. Can the authors include tests on longer horizon tasks like SWE bench and evaluate the effect of interruption and context change there?
2. One interesting setting could be that the original context/question contains error which is later corrected during the interruption/context change, and see if the model can recognize the error and reason with the correct change.

---

> ### Author Response · Authors · 2025-11-19
> **Rebuttal by Authors**
>
> We appreciate the reviewer for recognizing the contributions of the paper, including practical realistic problem setup and thorough investigation of the identified problem. We address the reviewer’s concerns below:
>
> ---
>
> **[W1, Q1] Justification of our settings on long-horizon tasks**
>
> We thank the reviewer for this suggestion and fully agree that the extension to other benchmarks, like SWE-bench, would be worth studying. That said, we believe that the results and the corresponding insights would be similar to those of the selected benchmarks. We would like to emphasize that our experimental setting is standardized and practically motivated as elaborated below.
>
> The math and coding benchmarks we use are not “short” in terms of model reasoning time: for example, LiveCodeBench problems in our setup often require more than 6 minutes of uninterrupted “thinking” on GPT-OSS, even on 80G A100 GPUs with vLLM speedups. These settings therefore naturally induce long reasoning traces where mid-trajectory interruptions are practically relevant. We also focus on such benchmarks because they are standard across SOTA reasoning models, providing a familiar and comparable testbed.
>
> ---
>
> **[W2] More Models such as DeepSeek**
>
> Following reviewers' suggestions, we broadened the scope of our analysis and included the results of the DeepSeek model in Appendix F.2. Running the full DeepSeek-R1 (685B) is computationally infeasible in our setting, and the DeepSeek-R1-Qwen variants are based on and close to Qwen, which we already study at multiple scales. The remaining option is the DeepSeek-R1-Llama3 series; we report DeepSeek-R1-Llama3-8b and DeepSeek-R1-Llama3-70B results in the Appendix F.2 of the revised paper.
>
> Regarding closed-source models, our interruption protocol requires mid-thinking control over the model’s internal reasoning stream (e.g., truncating or updating the ongoing chain of thought at specified token positions). Current APIs for systems such as o3 and Gemini 2.5 Pro expose only turn-level interfaces: the user provides a prompt, the model returns a completed answer, and there is no supported mechanism to deterministically cut off or edit the intermediate reasoning trace at a target step.
>
> ---
>
> **[W3] Result presentation**
>
> We appreciate the reviewer’s suggestions on using a single number to summarize the insights. That said, our main figures use line plots so readers can observe *how* metrics change as a function of interrupt position (e.g., late interruptions causing larger drops for update-driven interruptions, and the severity of reasoning leakage in early hard-interrupt settings).
>
> ---
>
> **[Q2] Error-correction / context-correction scenarios**
>
> We agree this is a important setting to study. Under the update-driven interruption settings, we do already instantiate a related scenario where the model begins reasoning under an initially flawed or incomplete context and then receives a corrective update mid-trajectory. A canonical example is modifying the starter code in coding tasks (Appendix E), where the “update” fixes earlier starter code error, clarifies constraints, and/or provides additional test cases.
>
> ---
>
> ###

---

### Official Review · Reviewer_Tdaj · 2025-11-01

**Soundness:** 2
**Presentation:** 3
**Contribution:** 2
**Rating:** 2
**Confidence:** 4

**Summary:**

This work explores how large reasoning models respond to interruption scenarios. Specifically, the authors argue that current large models generate responses in a turn-based static manner, which does not align with the continuous interaction demands faced by agents in modern language model applications. To address this, the authors propose a novel framework and benchmark for evaluating model performance under interrupted conditions, and summarize several common failure patterns observed in models.

**Strengths:**

1. This paper presents a highly novel perspective by exploring critical challenges that future language models may encounter in environment interaction scenarios. The proposed evaluation framework and benchmark establish a foundation for assessing agent capabilities, potentially contributing to the development of autonomous agents in future research.

2. The study provides reasonably comprehensive experiments with persuasive results. Particularly, the systematic categorization of failure patterns offers valuable insights into fundamental limitations of current language models.

**Weaknesses:**

1. While the authors propose that models require interruption handling in agent scenarios, the selected benchmarks (GSM8K, MATH500, AIME 2024/2025, and LiveCodeBench v6) do not inherently involve agent-based tasks. This raises questions regarding the motivation and applicability of their proposed framework. From a language modeling perspective, both hard and soft interruptions disrupt the causal nature of language generation. If feasible, incorporating tasks that explicitly require agent-like behavior would help substantiate the motivation and contextual relevance of their approach.

2. The strategy of selecting interruption points proportionally appears suboptimal. A more principled approach would be to treat individual sentences as reasoning units and use them as logical boundaries for inserting interruptions.

3. The design of the update mechanism remains unclear in its validity. By maintaining the original problem statement in the augmented problem with updates, the evaluation may simply encourage the model to "memorize" the base benchmark rather than genuinely assessing its ability to resolve interruptions in complex, generalized scenarios.

**Questions:**

1. Could the evaluation framework and benchmarks be extended to include more agent-oriented datasets? This would significantly strengthen the rationale for studying model interruption in practical scenarios.

2. Please elaborate on the advantages of interrupting language models compared to alternative approaches such as backtracking or turn-based methods.

3. Are there more principled and dynamic methods for interrupting models? The current approach, while valuable, might still be considered somewhat "frozen" in its implementation.

---

> ### Author Response · Authors · 2025-11-19
> **Rebuttal by Authors**
>
> Thanks for recognizing the contribution of our work, including the novel problem setup and evaluation framework, as well as the comprehensive experiments and results. Below, we would like to address the reviewer’s concerns:
>
> ---
>
> **[W1, Q1] Task Selection: From Foundational Math/Coding Towards Agentic Tasks**
>
> Our main goal and contribution is to provide a foundational, controlled study of interruptions in large foundation models. We thus start from math and coding tasks with reasoning models because (i) unlike dialogue, these benchmarks induce long “thinking” traces that make mid-reasoning interruption practically motivated (e.g., LiveCodeBench problem takes more than 6 minutes on average for GPT-OSS to complete even on 80G A100 GPUs coupled with vLLM speedup), (ii) they are standard across SOTA reasoning models and thus provide a familiar, comparable testbed, and (iii) “time” can be measured and reproducibly controlled internally via the number of thinking tokens, avoiding confounds such as API latency or environment stochasticity in full agentic settings. In this controlled yet realistic setting, we conduct a deep dive over four tasks, three model families, and three interruption scenarios, identify three recurring failure modes, and provide ablations (e.g., model size, interruption mechanism) that we intend to serve as a solid basis for future work. The framework is compositional and naturally extends to additional benchmarks (e.g., agentic tasks) and richer multi-turn dynamics (for example, update → soft interrupt → update). We agree that extending the framework to more agent-oriented datasets is an important next step (as mentioned in the Limitations section in the original version). We appreciate the reviewer's suggestion and we now state this more explicitly in the revised Limitations section.
>
> ---
>
> **[W1, Q2] Clarification on Autoregressive Causality, Backtracking, and Turn-based Methods**
>
> Thanks for bringing up this issue. We agree that language generation is causal in nature. To embrace this feature, we interrupt the model by inserting new tokens rather than backtracking (i.e., deleting) the already generated tokens. Yet, we do discuss the backtracking (i.e., restart from scratch) in lines 428–431 of the submission where the result is equivalent to that of the “Full Thinking Oracle”; we observe that restarting from scratch is on average computationally more expensive. In addition to interrupting models’ thinking trace, we also ablated the turn-based method, where the update instructions are inserted into the user’s turn (the results are reported in Section F.1 of the submission).
>
> ---
>
> **[W2] Interruption at Sentence Boundaries**
>
> Thank you for the suggestion. We agree that letting the model finish its current thought and interrupting at sentence-level boundaries would be an interesting ablation. Following your suggestion, we ablated this interruption methodology in the revised version of the paper using Qwen3-8B on math benchmarks. Specifically, instead of cutting at the proportional token position, we wait for the model to finish its current reasoning block (i.e., the next new line `\n`) and then interrupt. As shown in the updated Appendix F.3, this sentence-based method yields almost identical results to our proportional token-based method under hard and update-driven interrupt scenarios.
>
> ---
>
> **[W3] Checking Validity and Eliminating “Memorization”**
>
> To validate our benchmark and eliminate the possibility of benchmark memorization, we run the models on the augmented/revised problems as inputs without any interruptions (i.e., update instructions). With this setup when evaluated against the ground truth answers of the *original problems*, the models have nearly zero performance, indicating that the models are solving the revised problems, rather than than memorizing the specifications and/or answers of the original problems.
>
> ---
>
> **[Q3] Interrupt Mechanism and Alternatives**
>
> To isolate and analyze the models’ behavior, we studied several interrupt positions and three interrupt types as variables, keeping the rest of the environment fixed. Specifically, in our setup, we inject new or corrected information while the model has already been “thinking” for tens of seconds to minutes. This breaks the conventional frozen-input evaluation setup (single static prompt → one-shot answer) and allows us to systematically study how models handle being interrupted mid-reasoning. We agree that adding stochasticity to the interruption process itself is a reasonable next step for future work, which we acknowledge in the revised Limitations section.

---

### Official Review · Reviewer_Xkrr · 2025-11-01

**Soundness:** 3
**Presentation:** 3
**Contribution:** 3
**Rating:** 4
**Confidence:** 4

**Summary:**

This paper explores an unusual setting where language model generations are interrupted in the middle of chain of thought.

### **Novel Dynamic Evaluation Scenarios**
1. Interruptions: Test partial output quality under limited computation budget.
2. Dynamic Context: Assess adaptation to changes occurring mid-reasoning.

The paper introduces a novel dataset and evaluation benchmark to assess in math and coding domain, testing whether language models are interruptible by simple stop signs and correctly interpret newly introduced content from the world. They explore soft and hard stopping for the model, and the respective effectiveness of each interruption scenario.

### **Key Findings on Benchmarks**
They found out that reasoning models are able to gracefully end their thinking regardless of the model type and sizes. Moreover, to information update scenarios, most LRMs exhibit self-doubt when provided with additional information from the user's side. Even top LRMs (high accuracy in ideal conditions) suffer severe drops (up to 60%) when interrupted or facing late context updates in math/programming tasks requiring extended reasoning.

They identify three modes of reasoning failures:
- Reasoning Leakage: Models embed partial reasoning into final answers when interrupted.
- Self-Doubt: Performance worsens while trying to integrate update information.
- Panic: Under pressure, models skip reasoning and output wrong answers.

**Strengths:**

### **Novelty of Dynamic Evaluation for LRMs**
This paper suggest a important and novel direction of evaluation problem of reasoning models.
The dynamic environment setting for reasoning model seems unusual, but the provided examples below are reasonable.
1. User interruption for partial answers: In long-running tasks (e.g., proving a math theorem or debugging code), users often want progress updates or to abort early if the direction is wrong. Waiting hours for a final answer is unacceptable in interactive workflows.
2. Mid-task context updates: In collaborative coding (e.g., GitHub + AI agents), the codebase evolves while the model is thinking. A model that can't adapt to a new commit or user clarification is brittle.
3. Multi-agent environments: Future AI systems (e.g., coding agents in a shared repo) must handle concurrent changes—restarting from scratch breaks coordination.

> This dataset reflects the real-world demands as LRMs move into interactive, long-running, and collaborative roles.

### **Thorough analysis on Mid-reasoning Interruption**
1. Accuracy increases the later the interruption occurs in the reasoning trace.
- Models exhibit progressive reasoning - early interruptions yield poor results; late ones approach full performance.
- Magistral on coding tasks slightly improves when interrupted late (possibly due to overthinking avoidance).
2. Given soft interruption like end the reasoning faster, some model gives immediate wrong answer on hard tasks
3. Forcing to stop reasoning makes the model continue reasoning even in the output (e.g. \\boxed{} in math)

This shows that the models are hard to interrupt, either in soft or hard user instruction, and this implies that current reasoning models are hard to be deployed in interactive settings.

### **Analysis on model behavior when provided with extra context mid-reasoning**
1. Late-stage context updates cause sharp performance drops; models fail to recover or adapt reasoning trace.
2. Self-doubt causes models to ignore updates and stick to original reasoning, even when warned.
3. Prompt guidance (user-verified update tag) boosts adaptation, especially in math; coding remains robust.
4. Efficiency gain: Guided models recover with <110% token cost vs. full restart.
5. Late-stage updates still challenging on hard tasks (e.g., AIME).

**Weaknesses:**

### **Scope of Dataset Composition**

Although the paper introduces a novel evaluation direction, its dataset construction and analysis yield only incremental insights. It highlights compelling real-world scenarios requiring dynamic robustness in LRMs—such as multi-agent debate, mid-task context updates from agentic systems, and user interruptions—yet the dataset simulates only single-turn interruptions during model reasoning. The impact would be significantly greater with generalization to multi-turn, diverse dynamic scenarios.
Additionally, the authors should report dataset statistics, including:

- Distribution of interruption types,
- Total dataset size,
- Per-domain breakdown.

### **Lack of Diverse Model Analysis**

Furthermore, evaluations are limited to Qwen3 (1.7B/8B/32B), GPT-oss, and Magistral—a narrow set that does not reflect the broader landscape of open-source or closed-source LRMs. Testing widely used frontier models (e.g., DeepSeek-R1, o3, Claude 3.5, Gemini 1.5 Pro) is essential to substantiate claims of universal weakness in interrupt adaptation.Fast

**Questions:**

Please answer to the weakness part.

---

> ### Author Response · Authors · 2025-11-19
> **Rebuttal by Authors**
>
> We appreciate the reviewer for highlighting our paper’s contributions, including the new dynamic evaluation setting and thorough analyses. Below, we address each of the concerns:
>
> ---
>
> **[W1] Scope of Dataset Composition**
>
> For dataset construction and statistics, Appendix E in both the submission and the updated version details the augmentation steps and provides concrete examples. For clarity, we summarize the main statistics below. We run all interrupt types (hard interrupt variants, soft interrupt, and update-driven interrupt variants) on the full set of problems.
>
> | Benchmark | Size | Domain | Update Type(s) |
> | --- | --- | --- | --- |
> | GSM8K | 500 (subsampled) | Grade-school math | Variable changes that modify numeric values or conditions and change the final answer |
> | MATH-500 | 500 (original split) | Competition-style math | Variable changes that modify assumptions or parameters and change the final answer |
> | AIME 2024/2025 | 30 problems × 16 runs | High-level competition math | Variable changes applied to AIME-style questions (same inference protocol as DeepSeek-R1) |
> | LiveCodeBench v6 | 341 tasks (subset after Oct 1, 2024) | Coding problems | Starter-code updates, specification changes, and test-case updates that change behavior |
>
> In terms of the extension to more settings, we would like to emphasize that our main goal and contribution is to provide a foundational, controlled study of interruptions in large foundation models. We start from math and coding tasks with reasoning models because (i) unlike dialogue, these benchmarks induce long “thinking” traces that make mid-reasoning interruption practically motivated (e.g., LiveCodeBench problem takes more than 6 minutes on average for GPT-OSS to complete even on 80G A100 GPUs coupled with vLLM speedup), (ii) they are standard across SOTA reasoning models and thus provide a familiar, comparable testbed, and (iii) “time” can be measured and reproducibly controlled internally via the number of thinking tokens, avoiding confounds such as API latency or environment stochasticity in full agentic settings. In this controlled yet realistic setting, we conduct a deep dive over four tasks, three model families, and three interruption scenarios, identify three recurring failure modes, and provide ablations (e.g., model size, interruption mechanism) that we intend to serve as a solid basis for future work. The framework is compositional and naturally extends to additional benchmarks (e.g., agentic tasks) and richer multi-turn dynamics (for example, update → soft interrupt → update). We agree that extending the framework to more agent-oriented datasets is an important next step (as mentioned in the Limitations section in the original version). We appreciate the reviewer's suggestion and we now state this more explicitly in the revised Limitations section.
>
> ---
>
> **[W2] Lack of Diverse Model Analyses**
>
> We do our best in the paper to produce a representative set of models (three Qwen3 sizes, GPT-OSS, and Magistral), which represent five models at the forefront of current LRMs. In fact, as far as we are aware this set represents **all modern mid-sized unique open-source LRMs**, other open-source LRMs are either based on one of these models or too large to run (for example, Long-Cat is 675B parameters).
>
> Following reviewers' suggestions, we broadened the scope of our analysis and included the results of the DeepSeek model in Appendix F.2. Running the full DeepSeek-R1 (685B) is computationally infeasible in our setting, and the DeepSeek-R1-Qwen variants are based on and close to Qwen, which we already study at multiple scales. The remaining option is the DeepSeek-R1-Llama3 series; we report DeepSeek-R1-Llama3-8b and DeepSeek-R1-Llama3-70B results in the Appendix F.2 of the revised paper.
>
> We agree that including frontier closed-source models would further strengthen our empirical results and insights. However, our interruption protocol requires mid-thinking control over the model’s internal reasoning stream (e.g., truncating or updating the ongoing chain of thought at specified token positions). Current APIs for systems such as o3 expose only turn-level interfaces: the user provides a prompt, the model returns a completed answer, and there is no supported mechanism to cut off or edit the intermediate reasoning trace. Additionally, Claude 3.5 and Gemini 1.5 Pro are not reasoning models, making them inapplicable in our study.

---

> > ### Comment · Reviewer_Xkrr · 2025-11-27
> >
> > The authors rebuttal that clarifies the interruption is needed in single-turn long CoT reasoning, and adding several models has addressed my concerns. I incline towards accepting this paper, therefore I raise the score.

---

### Author Response · Authors · 2025-11-23
**Official Comment by Authors**

We thank the reviewers for all valuable questions and feedback. We posted our full rebuttal and an updated PDF on Nov 18, including revisions to the Limitations section and Appendix F (with all updates marked in red).

We appreciate the discussion phase and are happy to address any further questions. Please let us know your thoughts. Thanks!

---

### Meta-Review · Area_Chair_GXs8 · 2026-01-06

**Summary:**

All reviewers acknowledge that this submission is studying an interesting problem setup where the inference step leads to long (both in terms of number of tokens and also actual runtime) reasoning chains, and the context itself changes within that time --- the authors present it as an "interruption" to the reasoning of an LLM. Within that setup, the submission presents a benchmark that studies the effect of interruptions for different forms of interruptions and different stages in the LLM reasoning. The main concerns shared by all reviewers are that
- **W1:** Problems considered in the benchmark are not those that induce long enough reasoning chains to meaningfully highlight the effect of reasoning interruptions. The benchmark problems appear to be too simplistic in terms of inducing long reasoning chains, and the interruptions are cover a very limited scope (possibly because of the problems considered).
  - The authors responded to this by emphasizing that these benchmark problems are based on standard problem sets in LLM reasoning evaluations that induce long "thinking traces", and provide a foundation for a controlled evaluation of reasoning interruptions.
  - In my opinion, this only partially address the concerns. It is true that the problem setup is novel and could be quite interesting, the benchmark and the evaluation results are not able to highlight this. For example, the results are very much in line with what one would expect in LLM evaluations:
    - (i) For interruptions regarding time-constraints, it is not surprising that performance degrades gracefully based on how late the time-constraint interrupts are provided as earlier interrupts imply larger distribution shifts from the reasoning traces the LLMs are fine-tuned on than later interrupts.
    - (ii) For update-driven interrupts, it is not surprising that earlier interrupts have better performance as they would represent smaller distribution shifts from the reasoning data the LLMs are fine-tuned on as they represent complete problems, while later interrupts would induce a much larger distribution shift and thus worse performance.
    - Thus, given the somewhat expected results, it is not clear whether the problem sets and interrupts considered are providing us with a new facet for LLM evaluation.
- **W2:** The manuscript considers a small set of LLM families, and do not consider various frontier models.
  - The authors responded to this by considering an additional model family, highlighting the same results hold. Furthermore, the authors highlight that the proposed benchmark requires a form of API that various closed-source frontier models do not provide, and thus cannot be considered for an evaluation such as the one proposed in this submission.
  - In my opinion, this again only partially addresses the issue. Unless we envision significant changes to the APIs for the proprietary models, I see this as an indication that there is limited value in studying such fine-grained interruptions as we will not be able to ever provide third-party evaluations for closed-source models (which are the ones getting the most use). It might be more critical to consider the form of interruptions that are practically viable.


I believe that the proposed problem setup is very interesting, but the benchmark and evaluations need more development. Thus I am recommending a rejection for this submission.

**Reviewer Concerns:**

Beyond the critical concerns mentioned in the above **Summary**, the following are some additional concerns raised by the reviewers:
- Reviewers Tdaj and w9LU mentioned that the choice of the interruption points may not be appropriate, and it would be more reasonable to consider interruptions at the end of sentences. The authors responded to this by providing novel results with the end of sentences as interruption points, and highlighted that the same conclusions hold even with this alternate form of interruption points. The authors also presented results for absolute token position based interrupts instead of relative token position based interrupts. I believe these new results address this concern.
- Reviewer Tdaj also questioned the validity of the problems considered for the update-driven interruptions, highlighting that just modifying the original problems such that the updates lead to the same answer as the original problems limited the scope of the evaluation, with the possibility that models performing well on the base benchmark might also perform well even with the interruptions, confounding the effect of the update-driven interruptions. The authors responded to this by presenting new empirical results that show that the model outputs for the modified problem (without any additional update-driven interrupts) do not match the ground-truth answers of the original problems. In my opinion, this only partially addresses the concern as memorization would only come into play with the update-driven interrupts as the "modified problem + updates == original problem", and thus allow the model to find a shortcut. Since the modified problem (with incomplete or incorrect information) is not the same as the original problem, it is not surprising that the model generates a different output. This needs a different analysis to show that the model is just not utilizing its strong performance on the base benchmark and actually utilizing the updates in the interrupts to appropriately modify its reasoning.
- Reviewer w9LU brought up connections to existing literature on token-budget control and inference-time adaptation. The authors responded to this by highlighting that the dynamic setup considered in this submission is different from what is considered in literature, with the salient difference being that fact that, in this submission, the models are explicitly not aware of the updates/interrupts/constraints beforehand and need to adapt dynamically. I believe that this discussion addresses the concern.

**Reviewer Scores:**

Reviewer Xkrr had already mentioned that they would update their current score of 4 (marginally below acceptance threshold) to a positive score as the author responses addressed all their concerns.


Reviewers 2ARY and w9LU are positive with their current scores of 6 (marginally above acceptance threshold), and their concerns were only partially addressed in my opinion (as discussed regarding the critical concerns in the **Summary** section). Thus it is possible that they might have raised their scores by one point.


Reviewer Tdaj scored this submission as a 2 (reject, not good enough), and their concerns were only partially addressed (both the critical concerns, and the concern regarding the design of problems for the update-driven interrupts). Thus I would imagine that the reviewer would raise their scores at most by 1.

---

### Decision · Program_Chairs · 2026-01-26

Reject